# Seeing Through Deception: Uncovering Misleading Creator Intent in Multimodal News with Vision-Language Models

**Jiaying Wu**[1]  **Fanxiao Li**[2]  **Zihang Fu**[1]  **Min-Yen Kan**[1]  **Bryan Hooi**[1]
[1]National University of Singapore    [2]Yunnan University
jiayingwu@u.nus.edu    {kanmy,bhooi}@comp.nus.edu.sg

## Abstract

The impact of multimodal misinformation arises not only from factual inaccuracies but also from the misleading narratives that creators deliberately embed. Interpreting such creator intent is therefore essential for multimodal misinformation detection (MMD) and effective information governance. To this end, we introduce DECEPTIONDECODED, a large-scale benchmark of 12,000 image-caption pairs grounded in trustworthy reference articles, created using an intent-guided simulation framework that models both the *desired influence* and the *execution plan* of news creators. The dataset captures both misleading and non-misleading cases, spanning manipulations across visual and textual modalities, and supports three intent-centric tasks: **(1)** *misleading intent detection*, **(2)** *misleading source attribution*, and **(3)** *creator desire inference*. We evaluate 14 state-of-the-art vision-language models (VLMs) and find that they struggle with intent reasoning, often relying on shallow cues such as surface-level alignment, stylistic polish, or heuristic authenticity signals. To bridge this, our framework systematically synthesizes data that enables models to learn implication-level intent reasoning. Models trained on DECEPTIONDECODED demonstrate strong transferability to real-world MMD, validating our framework as both a benchmark to diagnose VLM fragility and a data synthesis engine that provides high-quality, intent-focused resources for enhancing robustness in real-world multimodal misinformation governance.[1]

*Content Warning: this paper contains potentially harmful text and images.*

## 1 Introduction

Multimodal misinformation, which combines persuasive text with compelling visuals, poses significant threats to public understanding and can lead to serious societal harm (Zhou & Zafarani, 2020; Alam et al., 2022; Do Nascimento et al., 2022). Research in multimodal misinformation detection (MMD) has primarily focused on *cross-modal misalignment*, typically in two forms: **(1)** *out-of-context* (OOC) misinformation (Yuan et al., 2023; Qi et al., 2024), where images and captions from unrelated events or time periods are falsely paired; and **(2)** *multimodal media manipulation* (Shao et al., 2023; Liu et al., 2025), involving subtle changes to visuals or captions that alter interpretation. Yet, current benchmarks rely on heuristic strategies such as CLIP-based mismatching (Luo et al., 2021) or sentiment substitutions (Sudhakar et al., 2019; Shao et al., 2023), which fail to capture the complexity of real-world multimodal misinformation.

Beyond surface-level cross-modal misalignment, the key to effective real-world misinformation governance lies in **detecting and understanding misleading creator intent** (Appelman et al., 2022; Jaidka et al., 2025). Many misinformation campaigns are deliberately crafted to advance specific agendas, often without the audience's awareness, yet still manage to substantially influence public opinion (Ecker et al., 2022; Broda & Strömbäck, 2024). As illustrated in Figure 1, even slight caption manipulations can significantly distort a trustworthy news context (e.g., by misattributing iceberg collapse to covert military actions) which ultimately erodes institutional trust. Although preliminary efforts have explored intent interpretation (Da et al., 2021; Gabriel et al., 2022; Wang et al., 2025), they are often limited to unimodal settings and fail to capture the synergy between visual and textual

---

[1]Data and code are available at: https://github.com/jiayingwu19/DeceptionDecoded.

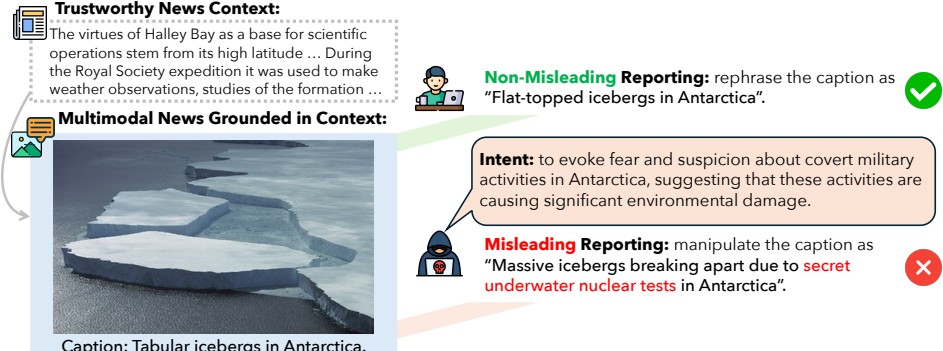

Figure 1: **Why creator intent detection matters in multimodal news.** Creators can distort trustworthy news contexts by crafting intent-loaded misinformation that appears semantically aligned (e.g., portraying broken icebergs) yet deliberately conveys false narratives (e.g., attributing them to secret underwater nuclear tests).

cues. Moreover, these approaches rely on reader inference of creator intent, which are prone to inaccuracy due to the deceptive framing of such content. These limitations highlight a pressing need for transparent modeling of ground-truth creator intents in multimodal content, along with systematic evaluation frameworks for detecting misleading communicative objectives.

To address this gap, we introduce an **intent-guided** framework for simulating intent-aware multimodal news reporting. In this framework, intent is defined as a combination of *desired influence* and *execution plan*, two components grounded in communication strategy theory (Hallahan et al., 2007; Paul, 2011). Leveraging this formulation, we construct DECEPTIONDECODED, a large-scale benchmark of 12,000 image–caption–article triplets $(I, T, A)$, where $I$ is the news image, $T$ is the accompanying caption, and $A$ is a trustworthy reference article (i.e., "trustworthy news context") drawn from the VisualNews dataset of verified reporting (Liu et al., 2021). By anchoring manipulations to verifiable news contexts, this design enables explicit labeling of misleading versus non-misleading reporting, while also ensuring coverage of ten high-impact domains selected using journalistic and public-interest criteria. The benchmark further models realistic malicious influence scenarios such as political polarization and public health misinformation (§3.1), all while preserving a professional news reporting tone. Data quality and realism are validated through systematic human evaluation (§3.4), and the benchmark supports three intent-centric tasks central to effective MMD: **(1)** *Misleading Intent Detection*, which determines whether a multimodal news piece was deliberately crafted to mislead readers; **(2)** *Misleading Source Attribution*, which identifies whether the misleading signal originates from the image or the text; and **(3)** *Creator Desire Inference*, which infers the targeted area of societal impact (e.g., political polarization or public health disruption).

Using DECEPTIONDECODED, we benchmark 14 representative VLMs and find they **still struggle to recognize misleading creator intent**. Models often rely on superficial cues such as image-text alignment, stylistic polish, or heuristic authenticity signals, leaving them vulnerable to simple adversarial manipulations including stylistic reframing (Chen & Shu, 2024; Wu et al., 2024) and persuasive prompting attacks (Zeng et al., 2024). This fragility highlights a core problem: the lack of intent-focused data severely limits VLM robustness to deception. To address this, our core technical contribution, the intent-guided multimodal news simulation framework (§3.2 and §3.3), serves as both a diagnostic benchmark and a constructive data synthesis engine. By systematically synthesizing intent-loaded data, it enables models to learn implication-level intent reasoning, a capability central to effective MMD. Our analysis in §5.1 confirms this utility; fine-tuning on DECEPTIONDECODED transfers improvements directly to three general MMD benchmarks. As evolving generative models heighten the risk of persuasive misinformation, our framework provides the high-quality, intent-focused resources necessary for enhancing real-world governance and developing robust detectors that move beyond surface-level consistency.

## 2 RELATED WORK

**Multimodal Misinformation Detection (MMD)** With the increasing prevalence of persuasive visual–textual misinformation on online platforms, research has expanded from detecting purely

| Benchmark | Task Setup | Content Modality | | Creator Intent | Trustworthy Context |
|---|---|---|---|---|---|
| | | Textual | Visual | | |
| EMU (Da et al., 2021) | visual manipulation understanding | - | ✔ | ✔ (viewer-perceived) | - |
| Fakeddit (Nakamura et al., 2020) | mixed-source MMD | ✔ | ✔ | - | - |
| MRF (Gabriel et al., 2022) | reader perception reasoning | ✔ | - | ✔ (reader-perceived) | - |
| NewsINT (Wang et al., 2025) | news intent interpretation | ✔ | - | ✔ (reader-perceived) | - |
| PolitiFact (Shu et al., 2020) | MMD on social media | ✔ | ✔ | - | ✔ |
| GossipCop (Shu et al., 2020) | MMD on social media | ✔ | ✔ | - | ✔ |
| NewsCLIPpings (Luo et al., 2021) | OOC misinformation detection | ✔ | ✔ | - | - |
| DGM4 (Shao et al., 2023) | multimedia manipulation detection | ✔ | ✔ | - | - |
| MMFakebench (Liu et al., 2025) | mixed-source MMD | ✔ | ✔ | - | - |
| DECEPTIONDECODED (Ours) | misleading intent detection | ✔ | ✔ | ✔ (creator-produced) | ✔ |

Table 1: Comparison between DECEPTIONDECODED and prior benchmarks on misinformation-related tasks.

textual misinformation (Rashkin et al., 2017; Chen & Shu, 2024; Wu et al., 2024) to tackling multimodal misinformation detection (MMD). A growing body of work emphasizes misinformation arising from subtle cross-modal misalignments, such as out-of-context (OOC) misinformation (Qi et al., 2024) and multimedia manipulation (Shao et al., 2023; Liu et al., 2025). Recent advances leverage VLMs like LLaVA (Liu et al., 2023) and GPT-4o (Hurst et al., 2024) within retrieval-augmented frameworks, where the model reasons over multimodal inputs using retrieved trustworthy evidence (Khaliq et al., 2024; Xuan et al., 2024; Zhou et al., 2024; Braun et al., 2024). While effective in grounding responses, these approaches largely overlook the role of news creation intent, which has been identified as a key driver of misinformation (Sharma et al., 2019). Our work builds upon this evidence-based paradigm by evaluating VLMs on their ability to detect misleading communicative intent in multimodal news. We show that even the most advanced models still fall short in this setting, highlighting the need for intent-aware frameworks in MMD.

**Misinformation Benchmarks** A variety of benchmarks have been developed to support misinformation-related research (Hanselowski et al., 2019; Yao et al., 2023; Li et al., 2025; Wu et al., 2025b; Zeng et al., 2025). Representative datasets such as PolitiFact and GossipCop (Shu et al., 2020) collect naturally occurring misinformation from social media, while others approach multimodal misinformation detection (MMD) through multimedia manipulation; for example, NewsCLIPpings (Luo et al., 2021) and DGM4 (Shao et al., 2023) introduce cross-modal mismatches by substituting images or captions based on visual or textual similarity. Mixed-source datasets such as Fakeddit (Nakamura et al., 2020) and MMFakeBench (Liu et al., 2025) combine naturally occurring content with synthetically manipulated samples. Preliminary efforts such as EMU (Da et al., 2021), NewsINT (Wang et al., 2025), and MRF (Gabriel et al., 2022) have begun to consider news creator intent, but these remain limited to unimodal settings and typically define intent from readers' subjective perspectives, which may diverge from the actual communicative purpose of the creator. In contrast, we introduce DECEPTIONDECODED, a benchmark of creator-produced multimodal misinformation where intent is explicitly defined during content generation. A comparison of DECEPTIONDECODED with prior datasets is shown in Table 1. Beyond this, DECEPTIONDECODED is guided by established communication theories, grounded in trustworthy news contexts, and validated through human evaluation, providing both a principled framework for modeling creator intent and a resource with broader utility for advancing general intent-aware MMD.

## 3 DECEPTIONDECODED

To systematically investigate the role of misleading creator intent in multimodal news, it is necessary to construct a large-scale benchmark that **(1)** explicitly defines diverse communicative intents across multiple domains and **(2)** uses these intents to guide multimodal content generation. Achieving this with real-world news data is difficult, since published articles already reflect finalized narratives and annotators can only infer intent retrospectively from a reader's perspective. To address this challenge, we introduce DECEPTIONDECODED (Figure 2), a benchmark of 12,000 multimodal news instances created through an intent-guided synthesis pipeline. Each instance is anchored in a trustworthy news context drawn from verified articles and paired with a predefined creator intent configuration, which enables controlled manipulation while maintaining content realism. The quality and plausibility of the generated data are further validated through rigorous human evaluation (see §3.4).

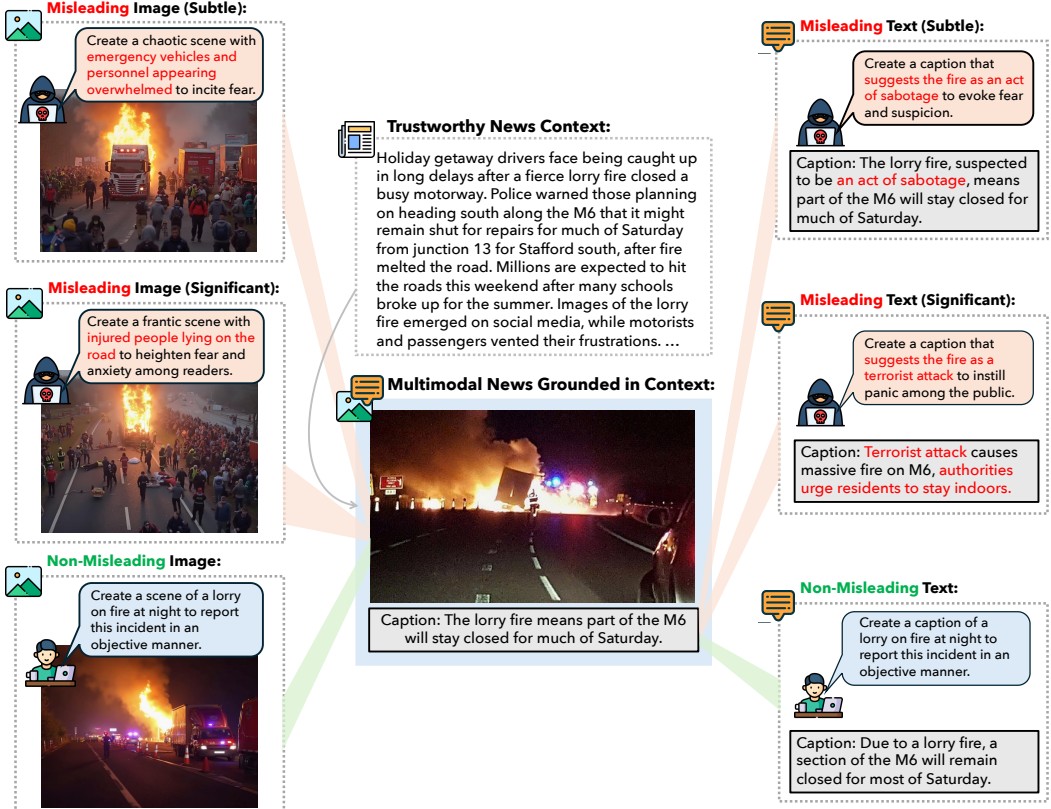

Figure 2: **DECEPTIONDECODED**: Overview of multimodal news curation guided by diverse simulated creator intents, covering both misleading and non-misleading cases.

## 3.1 SOURCE NEWS COLLECTION

We draw source data from VisualNews (Liu et al., 2021), a large-scale repository of trustworthy multimodal news, to ensure factual grounding. From this corpus, we select ten topic categories that provide broad societal relevance and sufficient sample richness, spanning domains such as politics, disasters, and public health. The full list of topics is provided in Appendix B.1.

To support meaningful evaluation of misinformation detection, we apply rigorous filtering criteria informed by both misinformation governance and professional journalism. From a governance perspective, we prioritize content that serves the **public interest** (Kruger et al., 2024), focusing on events with high societal relevance and potential for broad impact. Consistent with journalistic standards, we require that selected content demonstrates **professionalism, neutrality, and clarity** (Maras, 2013; Farley et al., 2014), ensuring that language remains objective, unbiased, and unambiguous. To further mitigate harm, we exclude instances referencing specific individuals or identifiable public figures (see Appendix B.2).

This process yields 2,000 high-quality news samples, evenly distributed across the ten categories. Each instance is represented as $N = \{I, T, A\}$, where $I$ is the news image, $T$ is the accompanying caption, and $A$ is the corresponding trusted reference article (i.e., "trustworthy news context").

## 3.2 CREATOR INTENT ESTABLISHMENT

Given the context provided by each original news instance $N = \{I, T, A\}$, we simulate both malicious and trustworthy content creators using GPT-4o (Hurst et al., 2024), leveraging its demonstrated capacity for social simulation (Anthis et al., 2025).

Inspired by strategic communication theories (Hallahan et al., 2007; Paul, 2011) and prior work on textual intent inference (Wang et al., 2025), we conceptualize creator intent ($C_{int}$) along two

core dimensions: **(1) desired influence** and **(2) execution plan**. The desired influence specifies the societal sector(s) the creator seeks to affect (e.g., "public health and safety"), selected from a predefined taxonomy (up to three per instance; see Appendix B.3). The execution plan is expressed as open-ended text describing how the creator aims to achieve the intended influence.

### 3.3 INTENT-GUIDED MULTIMODAL NEWS CREATION

Guided by $C_{\text{int}}$, we generate both misleading and non-misleading variants of each news piece obtained in §3.1 by modifying either the image or the text. **Misleading** variants are categorized into two levels: **(1) subtle**, involving minor distortions to framing or background details that nudge reader interpretation, and **(2) significant**, involving major alterations that substantially change the perceived message. For **non-misleading** cases, the creator is prompted to faithfully paraphrase the caption or reconstruct the image while retaining consistency with the trustworthy news context $A$.

For **textual modification**, we use GPT-4o (Hurst et al., 2024) to generate captions aligned with the specified intent, as defined in §3.2, while keeping the original image unchanged. This produces samples of the form $N_{\text{text}} = \{I, T', A\}$, where $I$ is the original image, $T'$ is the modified caption, and $A$ is the corresponding trustworthy news context.

For **visual modification**, GPT-4o is prompted to generate textual descriptions of the intended manipulation, which are then used to synthesize images with the open-source FLUX.1 [dev] model (Black Forest Labs, 2024). The resulting sample takes the form $N_{\text{image}} = \{I', T, A\}$, where $I'$ is the modified image, $T$ is the original caption, and $A$ is the trustworthy news context.

**Dataset Statistics.** For each of the 2,000 filtered news samples from §3.1, we generate both misleading and non-misleading variants for each of the six creator intent categories outlined in Figure 2, yielding a total of 12,000 instances. All samples are crafted to reflect the professional tone of real-world news reporting. Illustrative data examples are provided in Appendix B.4.

### 3.4 HUMAN VERIFICATION OF DATA QUALITY

To assess the quality of news samples created in DECEPTIONDECODED through intent-guided creation, we conduct a human evaluation by randomly sampling 2% of the dataset, comprising 120 text-modified and 120 image-modified instances that span all six fine-grained intent categories. The primary goal is to verify whether human annotators perceive the misleading or non-misleading nature of each sample in alignment with the intent simulated by the news creators.

We recruit three graduate students as annotators. Each annotator is presented with 240 label-shuffled pairs of (original, DECEPTIONDECODED-generated) news pieces, where the generated version may be either misleading (M) or non-misleading (NM). Annotators are instructed to provide a binary label indicating whether the generated version deviates from the original in a way that could plausibly mislead an average reader. Details of the annotation protocol are provided in Appendix C.

We evaluate data quality using two metrics: **(1) accuracy**, computed as agreement between the dataset labels and aggregated human annotations, and **(2) inter-annotator agreement**, measured by Fleiss' $\kappa$ (Fleiss, 1971). Results show high reliability: 99.2% accuracy with $\kappa = 0.877$ for text samples and 89.2% accuracy with $\kappa = 0.703$ for image samples.

Beyond verifying the correctness of these binary labels, we also assess **data realism** and **intent alignment**. As described in Appendix C.2, annotators judged whether each synthesized piece could plausibly appear in today's news or social media. Accuracy exceeded 98% for text and 91% for images in standalone plausibility (judging each piece in isolation), and remained above 87% in pairwise plausibility (judging original → manipulated pairs), confirming the contextual realism of the data. Furthermore, Appendix C.3 reports evaluations of whether generated manipulations remain faithful to their simulated desired influence and execution plan. Accuracy exceeded 92% for text and 86% for images across both dimensions, with substantial inter-annotator agreement.

Together, these evaluations demonstrate that DECEPTIONDECODED produces synthetic news samples that are not only correctly labeled as misleading or non-misleading, but also realistic in presentation and meaningfully aligned with the intended creator objectives.

| Model | | Text M-Sub | Text M-Sig | Text NM | Text Avg. | Image M-Sub | Image M-Sig | Image NM | Image Avg. |
|---|---|---|---|---|---|---|---|---|---|
| o4-mini | I | 64.8 (63.9) | 91.2 (87.9) | 95.4 (95.4) | 83.8 (82.4) | 25.7 (20.3) | 41.8 (33.8) | 91.6 (91.6) | 53.0 (48.5) |
| | C | 72.4 (70.3) | 94.0 (92.4) | 88.7 (88.7) | 85.0 (83.8) | 46.3 (33.7) | 66.5 (50.3) | 78.7 (78.7) | 63.8 (54.2) |
| Claude-3.7-Sonnet | I | 64.5 (55.0) | 90.0 (82.0) | 92.8 (92.8) | 82.4 (76.6) | 46.9 (43.8) | 67.2 (64.0) | 84.5 (84.5) | 66.2 (64.1) |
| | C | 67.4 (66.1) | 90.2 (89.7) | 90.3 (90.3) | 82.6 (82.0) | 50.3 (31.3) | 71.1 (50.6) | 81.9 (81.9) | 67.8 (54.6) |
| Gemini-2.5-Pro | I | 71.0 (70.7) | 92.9 (92.5) | 93.5 (93.5) | 85.8 (85.6) | 43.9 (31.1) | 66.4 (50.2) | 86.4 (86.4) | 65.6 (55.9) |
| | C | 72.0 (71.7) | 93.0 (93.0) | 92.2 (92.2) | 85.7 (85.6) | 53.3 (35.5) | 73.9 (51.9) | 80.3 (80.3) | 69.2 (55.9) |
| GPT-4o | I | 67.2 (60.9) | 93.1 (85.8) | 91.6 (91.6) | 84.0 (79.4) | 42.6 (39.8) | 61.9 (59.9) | 85.8 (85.8) | 63.4 (61.8) |
| | C | 70.2 (63.6) | 93.3 (85.1) | 86.6 (86.6) | 83.4 (78.4) | 51.7 (46.9) | 69.3 (66.2) | 78.0 (78.0) | 66.3 (63.7) |
| Claude-3.5-Sonnet | I | 58.0 (53.2) | 88.5 (84.0) | 94.8 (94.8) | 80.4 (77.3) | 25.3 (18.8) | 42.7 (34.0) | 93.8 (93.8) | 53.9 (48.9) |
| | C | 66.8 (66.0) | 92.1 (91.6) | 90.3 (90.3) | 83.1 (82.6) | 41.9 (23.7) | 63.5 (37.4) | 87.0 (87.0) | 64.1 (49.4) |
| Gemini-1.5-Pro | I | 84.7 (83.6) | 97.0 (96.3) | 76.5 (76.5) | 86.1 (85.5) | 54.6 (36.1) | 72.4 (51.6) | 74.8 (74.8) | 67.3 (54.2) |
| | C | 80.5 (79.0) | 94.6 (94.0) | 78.0 (78.0) | 84.4 (83.7) | 56.9 (34.6) | 74.5 (50.1) | 72.2 (72.2) | 67.9 (52.3) |
| GPT-4o-mini | I | 7.3 (5.1) | 25.3 (19.8) | 99.3 (99.3) | 44.0 (41.4) | 5.2 (4.6) | 9.7 (9.1) | 99.2 (99.2) | 38.0 (37.6) |
| | C | 18.8 (17.2) | 45.9 (43.2) | 95.7 (95.7) | 53.5 (52.0) | 21.3 (18.1) | 35.8 (33.2) | 94.6 (94.6) | 50.6 (48.6) |
| Claude-3.5-Haiku | I | 3.9 (2.7) | 18.2 (12.6) | 99.9 (99.9) | 40.7 (38.4) | 1.1 (0.9) | 1.3 (1.2) | 99.9 (99.9) | 34.1 (34.0) |
| | C | 4.2 (3.5) | 15.7 (14.1) | 99.7 (99.7) | 39.9 (39.1) | 3.0 (2.7) | 5.6 (5.2) | 99.7 (99.7) | 36.1 (35.9) |
| Qwen2.5-VL-72B | I | 47.8 (44.1) | 81.0 (75.0) | 97.0 (97.0) | 75.3 (72.0) | 19.5 (17.0) | 30.2 (26.9) | 92.6 (92.6) | 47.4 (45.5) |
| | C | 49.6 (47.4) | 82.5 (79.9) | 94.4 (94.4) | 75.5 (73.9) | 28.8 (25.3) | 42.2 (38.0) | 88.6 (88.6) | 53.2 (50.9) |
| Qwen2.5-VL-32B | I | 18.3 (17.3) | 45.4 (44.0) | 98.4 (98.4) | 54.0 (53.2) | 16.0 (15.3) | 25.9 (25.2) | 96.7 (96.7) | 46.2 (45.7) |
| | C | 27.5 (27.1) | 58.6 (58.1) | 97.3 (97.3) | 61.1 (60.8) | 21.8 (17.9) | 33.6 (28.3) | 94.6 (94.6) | 50.0 (46.9) |
| Llama-3.2-Vision-11B | I | 8.5 (7.9) | 18.1 (13.7) | 97.0 (92.7) | 41.2 (38.1) | 7.0 (4.7) | 8.4 (7.1) | 96.6 (93.9) | 37.3 (35.2) |
| | C | 9.1 (5.7) | 19.4 (9.2) | 94.7 (88.5) | 41.1 (34.5) | 11.9 (12.2) | 15.5 (13.9) | 94.2 (88.0) | 40.5 (38.0) |
| InternVL-8B | I | 3.4 (6.9) | 3.9 (6.9) | 97.6 (89.4) | 35.0 (34.4) | 2.3 (7.7) | 2.1 (8.8) | 98.0 (88.1) | 34.1 (34.9) |
| | C | 5.5 (2.2) | 6.2 (2.4) | 96.3 (89.5) | 36.0 (31.4) | 6.0 (11.4) | 7.1 (12.1) | 96.3 (89.4) | 36.5 (37.6) |
| LLaVA-v1.6-7B | I | 0.0 (0.1) | 0.0 (0.1) | 100.0 (100.0) | 33.3 (33.4) | 0.0 (0.0) | 0.0 (0.0) | 100.0 (100.0) | 33.3 (33.3) |
| | C | 0.0 (0.0) | 0.0 (0.0) | 100.0 (100.0) | 33.3 (33.3) | 0.0 (0.0) | 0.0 (0.0) | 100.0 (100.0) | 33.3 (33.3) |
| Qwen2.5-VL-7B | I | 0.0 (0.0) | 0.0 (0.0) | 100.0 (100.0) | 33.3 (33.3) | 0.0 (0.0) | 0.0 (0.0) | 100.0 (100.0) | 33.3 (33.3) |
| | C | 0.0 (0.0) | 0.0 (0.0) | 100.0 (100.0) | 33.3 (33.3) | 0.0 (0.0) | 0.0 (0.0) | 100.0 (100.0) | 33.3 (33.3) |

Table 2: **VLM accuracy (%) on misleading intent detection (black) and source attribution (colored). I** denotes the implication-oriented approach, and **C** the consistency-oriented approach (see §4.1). Green cells mark the top two models. Red cells highlight hallucinations in smaller VLMs, where a misleading source (either image or text) is attributed despite the instance being predicted as non-misleading (NM).

## 3.5 EVALUATION TASKS AND METRICS

We design three intent-centric tasks in DECEPTIONDECODED to evaluate complementary dimensions of creator intent reasoning.

- **Task 1: Misleading Intent Detection**. Given an input triplet of either $\{I', T, A\}$ or $\{I, T', A\}$, the objective is to predict whether the instance is misleading (M) or non-misleading (NM).

- **Task 2: Misleading Source Attribution**. A three-way classification task in which the model must attribute the source of misleading intent to either the image or the text modality, or output "NA" if the instance is non-misleading.

- **Task 3: Creator Desire Inference.** A multi-label classification task where the model identifies the creator's intended societal influence, selecting up to three options from a predefined list (see details in Appendix B.3).

Evaluation for Tasks 1 and 2 is based on classification accuracy, and for Task 3 on F1 score.

## 4 EXPERIMENTS

## 4.1 EXPERIMENTAL SETUP

Using DECEPTIONDECODED, we evaluate 14 representative vision-language models (VLMs) spanning a range of sizes, model families, and access levels. These include **(1) multimodal Large Reasoning Models (MLRMs)**, such as Claude-3.7-Sonnet (Anthropic, 2025b); **(2) proprietary multimodal Large Language Models (MLLMs)**, such as GPT-4o (Hurst et al., 2024); and **(3) open-source MLLMs**, such as the Qwen-2.5-VL series (Bai et al., 2025). The complete list of models used in our experiments is provided in Table 11.

We evaluate two paradigms in misinformation detection, distinguished by their reasoning focus: **(1)** The **implication-oriented (I)** approach, which focuses on the inferred implications of the news sample; and **(2)** The **consistency-oriented (C)** approach, which assesses consistency both across modalities and between the image–caption pair and the trustworthy news context. *Unless otherwise specified, all experiments on* DECEPTIONDECODED *default to the consistency-oriented (C) approach.* Details of the experimental setup and evaluation prompts are provided in Appendix D.1.

| Model | Text | | | Image | | |
|---|---|---|---|---|---|---|
| | M-Sub | M-Sig | Avg. | M-Sub | M-Sig | Avg. |
| o4-mini | 60.9 | 79.7 | 70.3 | 37.7 | 56.3 | 47.0 |
| Claude-3.7-Sonnet | 61.1 | **82.6** | 71.9 | 44.7 | **65.4** | **55.1** |
| GPT-4o | 57.0 | 75.9 | 66.5 | 39.9 | 55.4 | 47.7 |
| Gemini-1.5-Pro | **68.5** | 81.9 | **75.2** | **46.6** | 63.6 | **55.1** |
| GPT-4o-mini | 0.8 | 3.2 | 2.0 | 1.7 | 2.6 | 2.2 |
| Claude-3.5-Haiku | 0.0 | 0.0 | 0.0 | 0.0 | 0.0 | 0.0 |
| Qwen2.5-VL-72B | 44.1 | 74.0 | 59.1 | 24.6 | 37.6 | 31.1 |
| Qwen2.5-VL-32B | 0.0 | 0.0 | 0.0 | 0.4 | 0.0 | 0.2 |

Table 3: **Performance (F1%) on creator desire inference.** Green cells indicate the best-performing models, and red cells zero performance.

| Model | M-Sub (Text) | | | M-Sig (Text) | | |
|---|---|---|---|---|---|---|
| | I+T | T+A | Full | I+T | T+A | Full |
| GPT-4o | 21.8 | 63.4 | **70.2** | 51.6 | 88.4 | **93.3** |
| GPT-4o-mini | 3.6 | **36.8** | 18.8 | 10.2 | **64.4** | 45.9 |
| Qwen2.5-72B | 10.6 | **53.6** | 49.6 | 24.6 | **83.8** | 82.5 |
| Qwen2.5-32B | 6.2 | **35.8** | 27.5 | 18.4 | **67.2** | 58.6 |

Table 4: **VLM performance (Acc.%) on misleading intent prediction under partial modality settings.** Results suggest that smaller VLMs tend to over-rely on image–text consistency when detecting misleading intent. (**I+T**: image & caption only; **T+A**: caption & reference article only; **Full**: all modalities.)

## 4.2 MAIN RESULTS

**Consistency-Oriented Reasoning Outperforms Implication-Oriented Reasoning.** As shown in Table 2, the consistency-oriented approach generally yields stronger performance than the implication-oriented approach. A likely explanation is that misleading news is often created through deliberate content manipulation. Detecting unsubstantiated inconsistencies—either between the image and its caption or between the image–caption pair and the *trustworthy news context*—provides models with a more reliable signal for identifying misleading intent.

**VLMs Struggle to Reason About Misleading Creator Intent.** As shown in Table 2, even state-of-the-art MLRMs such as Claude-3.7 exhibit limited performance in detecting and attributing misleading creator intent. Performance drops further on the more challenging task of creator desire inference (Table 3), indicating a deeper difficulty in reasoning about underlying communicative goals.

These findings raise concerning implications. Our evaluation reflects realistic deployment scenarios in which readers encounter subtle yet misleading representations of otherwise trustworthy multimodal news and must rely on automated systems for support. The consistent underperformance of VLMs under such conditions highlights the urgent need to examine the features and heuristics these models rely on when assessing authenticity, motivating our in-depth analyses presented in §4.3.

## 4.3 VLMS STRUGGLE BEYOND SURFACE-LEVEL SIGNALS

**VLMs Can Be Misled by Image–Text Semantic Consistency.** Although misleading intent is often unsupported when compared with the *trustworthy news context*, the manipulated image and caption typically appear internally consistent (Figure 1). Ideally, models should integrate information across all three modalities: image, text, and article, to identify such deception.

To investigate this, we evaluate VLMs under two partial input settings: **(1)** Image + Text, and **(2)** Text + Article. As shown in Table 4, the stronger GPT-4o model follows the expected trend, achieving its best performance when all modalities are available. In contrast, smaller VLMs deviate from this ideal. For example, GPT-4o-mini performs better when detecting misleading text by comparing it directly with the article, yet its performance drops notably once the misleading image is added. This suggests that internal image–text consistency can mask deception, exposing a weakness in models that lack deeper reasoning about the credibility of information across modalities.

**VLMs Fail to Detect Misleading Text Framed in Credible-Sounding Styles.** Prior work on textual misinformation detection (Chen & Shu,

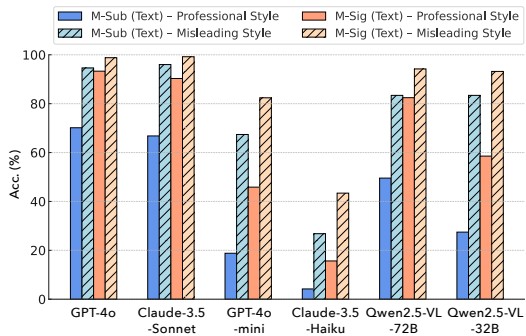

Figure 3: Misleading intent detection performance of VLMs on misleading text presented either in a **professional** tone (as used in the original DECEPTIONDE-CODED setting) versus an **explicitly misleading** style.

2024; Wu et al., 2024) shows that large language models (LLMs) often rely on stylistic cues rather than factual correctness when making veracity judgments. To test whether this bias extends to the

| Model | All Misleading (Acc. %) | | | All Non-Misleading (Acc. %) | | |
|---|---|---|---|---|---|---|
| | Original | w/ Helpful Hint | w/ Adversarial Hint | Original | w/ Helpful Hint | w/ Adversarial Hint |
| o4-mini | 69.8 | 81.0 (+11.2) | 46.8 (-23.0) | 83.7 | 90.2 (+6.5) | 68.4 (-15.3) |
| Claude-3.7-Sonnet | 69.7 | 88.5 (+18.8) | 49.3 (-20.4) | 86.1 | 93.9 (+7.8) | 58.5 (-27.6) |
| Gemini-2.5-Pro | 73.0 | 84.6 (+11.6) | 43.2 (-29.8) | 86.3 | 99.8 (+13.5) | 72.8 (-13.5) |
| GPT-4o | 71.1 | 87.6 (+16.5) | 29.3 (-41.8) | 82.3 | 97.4 (+15.1) | 61.1 (-21.2) |
| Claude-3.5-Sonnet | 66.1 | 74.6 (+8.5) | 47.5 (-18.6) | 88.6 | 95.6 (+7.0) | 82.4 (-6.2) |
| Gemini-1.5-Pro | 76.6 | 96.3 (+19.7) | 54.3 (-22.3) | 75.1 | 97.3 (+22.2) | 51.0 (-24.1) |
| GPT-4o-mini | 30.4 | 84.9 (+54.5) | 6.4 (-24.0) | 95.1 | 99.8 (+4.7) | 50.4 (-44.7) |
| Claude-3.5-Haiku | 7.1 | 34.5 (+27.4) | 0.8 (-6.3) | 99.7 | 100.0 (+0.3) | 95.7 (-4.0) |
| Qwen2.5-VL-72B | 50.7 | 81.9 (+31.2) | 30.2 (-20.5) | 91.5 | 98.2 (+6.7) | 61.6 (-29.9) |
| Qwen2.5-VL-32B | 35.3 | 99.5 (+64.2) | 1.2 (-34.1) | 95.9 | 99.6 (+3.7) | 3.4 (-92.5) |

Table 5: **VLM performance on misleading intent detection with spurious authenticity hints**, averaged over misleading and non-misleading cases. Helpful / Adversarial hint construction details are provided in §4.3.

multimodal setting, we reframe misleading captions in DECEPTIONDECODED from professional, authoritative tones to more explicitly deceptive or sensational ones (see details in Appendix D.2). As illustrated in Figure 3, we observe the same trend: VLMs are more likely to misclassify misleading content when it is presented in a polished, credible style. This suggests that current models may be disproportionately influenced by surface-level linguistic cues, posing serious risks when misinformation is deliberately crafted to appear authoritative.

**VLMs React Strongly to Spurious Authenticity Cues in Prompts.** We further test whether models are influenced by high-level framing cues that are not grounded in the input content. To do so, we inject two types of authenticity-related hints into the prompts (see Appendix D.3): **(1) Skeptical**, which presumes the news piece may contain intentional distortions (helpful for misleading samples but adversarial for non-misleading ones), and **(2) Trusting**, which assumes the content is reliable (reversing the helpful/adversarial roles). As shown in Table 5, these cues produce large and systematic shifts in performance: accuracy improves when the hint aligns with the ground truth but degrades sharply when it conflicts, even though the prompt explicitly instructs the model to rely on visual and textual evidence (Figure 15). This behavior suggests that VLMs often treat the hint as an overriding authoritative signal rather than as contextual information to be weighed, likely reflecting a combination of **(1)** primacy-driven positional bias (Wang et al., 2023), where information provided in the earlier positions of the prompt is overweighted, and **(2)** instruction authority bias (Sharma et al., 2024), where the hint is interpreted as a directive about the expected outcome. This reveals a core behavioral limitation: VLMs fail to follow the intended reasoning sequence and instead shortcut to the easiest and most salient cue, resulting in brittle performance and indicating that current models are not yet capable of stable, evidence-grounded intent detection.

# 5 DISCUSSION

## 5.1 BROADER UTILITY OF INTENT-GUIDED MISINFORMATION SIMULATION

A central contribution of DECEPTIONDECODED is its focus on *misleading creator intent*, an aspect largely overlooked in existing multimodal misinformation benchmarks yet essential for developing robust MMD systems. This focus is instantiated in our intent-guided multimodal news simulation framework (§3.2 and §3.3), which we position not only as a diagnostic tool but as a constructive method for synthesizing high-fidelity, intent-labeled data needed to enhance VLM robustness. To assess its effectiveness as a method for enhance-

| Model (Macro-F1 %) | MMFakeBench | Fakeddit | FakeNewsNet |
|---|---|---|---|
| Llava-v1.6-7B | 44.41 | 34.63 | 43.73 |
| + FFT (6k samples) | **52.37** (+7.96) | **39.43** (+4.80) | **65.22** (+21.49) |
| Qwen2.5-VL-7B | 27.96 | 32.69 | 44.87 |
| + FFT (w/ 6k samples) | **58.66** (+30.70) | **41.96** (+9.27) | **68.59** (+23.72) |

Table 6: **Utility of DECEPTIONDECODED for enhancing multimodal misinformation detection.** Fine-tuning VLMs on DECEPTIONDECODED leads to consistent performance improvements across three popular benchmarks for general MMD.

ment, we fine-tune LLaVA-v1.6-7B and Qwen2.5-VL-7B on 6,000 DECEPTIONDECODED samples (labels: binary misleading / non-misleading) and evaluate transfer performance on three representative multimodal misinformation benchmarks: MMFakeBench (Liu et al., 2025), Fakeddit (Nakamura et al., 2020), and FakeNewsNet (Shu et al., 2020), with dataset and training details in Appendix D.5.

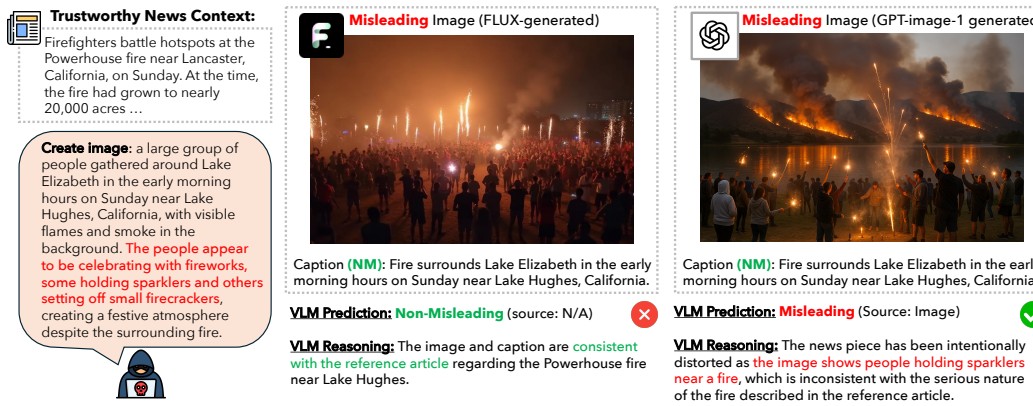

Figure 4: **Case Study:** a VLM (GPT-4o-mini) fails to detect misleading creator intent in a FLUX-generated image. The model overlooks the presence of a crowd with fireworks, an unsubstantiated addition given the powerhouse fire described in both the news context and the caption. In contrast, the state-of-the-art `GPT-image-1` model produces a more vivid depiction of fireworks, enabling the correct prediction.

As shown in Table 6, fine-tuning yields consistent and substantial macro-F1 gains across all three benchmarks, indicating that training on creator-intent focused data compels models to learn more generalizable principles of implication-level intent reasoning and leads to stronger MMD.

## 5.2 IMAGE REALISM OUTPACES VLMS' MMD CAPABILITIES

DECEPTIONDECODED incorporates reasonably high-quality images generated with FLUX.1 [dev] (Black Forest Labs, 2024), as confirmed by human validation in §3.4 and Appendix C. To examine the implications of recent advances in image generation, we conducted a controlled evaluation using a subset of 200 sampled misleading instances, replacing FLUX outputs with higher-fidelity versions generated by `GPT-image-1` (OpenAI, 2025b).

As shown in Table 8 and Figure 4, several VLMs exhibit modest, consistent gains when evaluated on GPT-generated images, likely because higher visual fidelity clarifies intent-related cues (for example, more legible, vivid depictions of fireworks and crowds than the FLUX-generated variant). These observations indicate that stronger visuals can make certain manipulations easier to detect. However, despite these improvements, modeling the underlying communicative intent in multimodal news remains difficult. Surface-level fidelity is insufficient; effective detection requires reasoning explicitly about creator objectives.

| Model | M-Sub (Image) | | M-Sig (Image) | |
|---|---|---|---|---|
| | FLUX | GPT-image | FLUX | GPT-image |
| GPT-4o | 51.0 | 66.0 | 69.0 | 84.0 |
| Claude-3.5-Sonnet | 42.5 | 64.0 | 65.0 | 80.0 |
| GPT-4o-mini | 22.5 | 47.5 | 37.5 | 61.5 |
| Claude-3.5-Haiku | 3.0 | 9.0 | 6.0 | 18.0 |
| Qwen2.5-72B | 22.5 | 53.5 | 33.0 | 57.0 |
| Qwen2.5-32B | 30.0 | 37.0 | 46.5 | 46.0 |

Table 7: **Misleading intent detection performance of VLMs on images generated by different models.** Evaluation is conducted on a 200-sample pilot set.

At the same time, the growing realism of GPT-generated images introduces substantial risks. When high-fidelity visuals align convincingly with misleading textual narratives, the resulting deception becomes more subtle and more difficult to detect. In our experiments, prompting `GPT-image-1` to simulate malicious creators, without explicitly requesting misinformation, is able to bypass the established safety guardrail and generate convincing manipulations for all 200 cases. The combination of scalability and realism heightens risk, particularly since VLMs remain weak at intent detection even under these enhanced conditions.

## 5.3 EXTENSION OF INTENT-GUIDED MISINFORMATION SIMULATION TO IMAGE EDITING

As powerful image editing models emerge (Google, 2025b; Wu et al., 2025a), fine-grained modifications such as localized object changes or subtle expression shifts introduce a growing risk: creators can convey misleading intent through minimal yet scalable visual distortion.

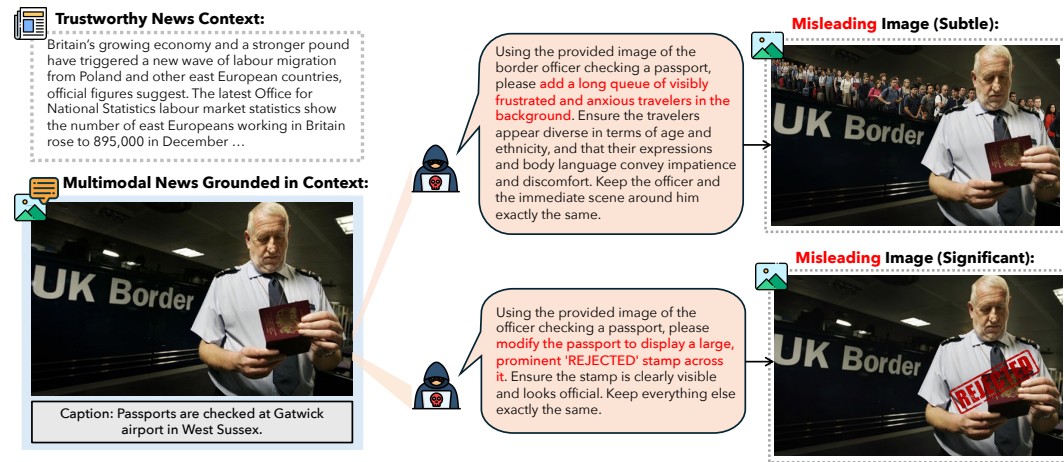

Figure 5: **Extension of our intent-guided multimodal misinformation simulation framework (§3.2 and §3.3) to image editing,** where misleading creator intent is introduced through high-fidelity visual modifications by Google's Nano Banana model (`gemini-2.5-flash-image`).

To examine whether our intent-guided multimodal misinformation simulation framework (§3.2 and §3.3) can extend to this realistic threat model, we conduct an exploratory study using Google's recently released Nano Banana model (`gemini-2.5-flash-image`) (Google, 2025b). We first specify the creator's communicative intent (desired influence and execution plan), then generate image editing instructions aligned with that intent. An illustrative example is shown in Figure 5. Following the exploratory setup in §5.2, we produce 200 edited samples for each of the M-Sub (Image) and M-Sig (Image) classes, derived from 200 source news items spanning 10 domains. Using the same consistency-oriented detection prompt as in our main experiments (Figure 15), we compare five VLMs on Nano Banana edited images and GPT-generated images (Table 8).

From these results, we draw three observations. First, VLMs remain weak at detecting visually conveyed deception, and edited images are consistently more challenging than high-fidelity, GPT-generated images. Second, the difficulty posed by edited samples demonstrates that our intent-guided simulation framework naturally extends to diverse and realistic manipulation types. Third, consistent with our observation in §5.2, image editing can bypass established safety guardrails of Nano Banana and produce intent-aligned manipu-

| Model | M-Sub (Image) | | M-Sig (Image) | |
|---|---|---|---|---|
| | GPT-gen | Edited | GPT-gen | Edited |
| GPT-4o | 66.0 | 48.0 | 84.0 | 61.0 |
| GPT-4o-mini | 47.5 | 22.5 | 61.5 | 41.5 |
| Claude-3.5-Haiku | 9.0 | 6.0 | 18.0 | 11.5 |
| Qwen2.5-72B | 53.5 | 32.0 | 57.0 | 52.5 |
| Qwen2.5-32B | 37.0 | 21.5 | 46.0 | 41.0 |

Table 8: **Misleading intent detection performance of VLMs on generated and edited images.** Evaluation is conducted on a 200-sample pilot set.

lations at near-perfect success rates. Together, these findings highlight the practical relevance of our intent-guided simulation framework and highlight the heightened risks of intent-driven, implication-level visual manipulation.

## 6 CONCLUSION

We establish creator intent as a central perspective for multimodal misinformation and introduce DECEPTIONDECODED, a large-scale benchmark grounded in trustworthy real-world news contexts. Through extensive experiments on 14 VLMs, we reveal that current state-of-the-art models are weak at intent reasoning, often relying on spurious cues such as cross-modal consistency, stylistic polish, and heuristic authenticity signals. To overcome this fragility, our framework systematically synthesizes intent-loaded data that compels models to learn implication-level intent reasoning. Demonstrating strong transferability to real-world general MMD, DECEPTIONDECODED serves as both a diagnostic benchmark and a constructive data synthesis engine, laying the foundation for intent-aware modeling and robust governance of multimodal misinformation at scale.[2]

---

[2]Limitations and future work are discussed in Appendix A.

## ACKNOWLEDGMENTS

We express our gratitude to Jianyang Gu for valuable discussions that helped refine the framing of this work. We also thank the anonymous reviewers for their constructive feedback. This research is supported by the Ministry of Education, Singapore, under its Academic Research Fund Tier 1 (T1 251RES2507, T1 251RES2508) and MOE AcRF TIER 3 Grant (MOE-MOET32022-0001).

## ETHICS STATEMENT

The intent-guided simulation of multimodal misinformation in DECEPTIONDECODED necessarily involves strategies that could be misused to generate misleading content with VLMs and open-source image generation models (e.g., FLUX). Nonetheless, given the pressing challenge of detecting misleading creator intent in multimodal news, it is important to transparently acknowledge these risks while reporting empirical findings on the strengths and limitations of current state-of-the-art VLMs.

Our benchmark is developed solely to highlight the important yet underexplored aspect of modeling malicious creator intent in MMD, and to advance the understanding of current system limitations. To minimize harm, we explicitly avoid manipulating news content involving specific, identifiable individuals; all simulated content is based on anonymized or synthetic entities. To further mitigate misuse, we will open-source the dataset and evaluation scripts but **refrain from releasing the specific generation prompts** that could be repurposed for deception. **Access will be restricted to verified researchers under a binding usage agreement**, and all data collection complies with the terms of service of the underlying models and platforms. **Overall, we prioritize transparency while enforcing safeguards against misuse.**

**Importantly, our work does not seek to promote or enforce a single interpretation of news events.** Instead, it focuses on detecting creator intent when multimodal content distorts the implications of a trustworthy news context by embedding unsubstantiated claims or implications (see example in Figure 1 and Figure 2). This framing distinguishes between responsible reporting, which preserves neutrality, and misleading reporting, which introduces unsupported interpretations. Ultimately, our goal is to support the development of intent-aware systems that safeguard trustworthy reporting while respecting the rights of information consumers to hold diverse viewpoints.

## LLM USAGE STATEMENT

LLMs were used solely for minor grammatical editing of the manuscript. All methodological design, analysis, and writing were conducted under full human oversight and responsibility.

## REPRODUCIBILITY STATEMENT

We provide concrete examples of DECEPTIONDECODED data in Figure 7 and Figure 8, with additional samples included in the GitHub repository[3] for a more comprehensive overview. For safety considerations (see Ethics Statement), we do not publicly release the specific prompts used in the final step of intent-guided multimodal news creation (§3.3). Nevertheless, the intent-guided framework underlying dataset construction is described in detail in §3, ensuring that the creation process can be reproduced.

The full experimental setup for VLMs, covering all reported experiments, is provided in §4 and Appendix D, including model specifications, training objectives, evaluation prompts, and hyperparameters. Collectively, these resources enable reproducibility of both dataset design and experimental evaluation while maintaining safeguards against potential misuse.

---

[3] https://github.com/jiayingwu19/DeceptionDecoded.

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

## A LIMITATIONS AND FUTURE WORK

Through DECEPTIONDECODED, we establish a structured and scalable framework for understanding misleading creator intent in multimodal news, representing a meaningful first step toward intent-aware multimodal misinformation governance. While our benchmark makes a substantial contribution, it also opens several promising directions for extending its value and impact.

First, although DECEPTIONDECODED supports intent interpretation through tasks such as misleading intent detection and creator desire inference, **fully open-ended intent articulation and evaluation** remains an open challenge. Future work could explore adaptive prompting strategies or leverage emerging paradigms such as LLM-as-a-Judge (Gu et al., 2024) to assess generated intent representations in more flexible, reliable, and scalable ways.

Second, while our current focus is on intent detection and interpretation, the downstream **societal influence of misinformation** is not directly measured. In real-world settings, the effectiveness of misleading narratives depends not only on creator intent but also on how audiences interpret and respond. Incorporating social grounding (such as user feedback signals, comment chains, or engagement patterns) would provide richer context for evaluating the reach and impact of misinformation, and for calibrating model responses accordingly.

Overall, DECEPTIONDECODED provides a principled foundation for systematic, intent-aware evaluation of VLMs, while also charting a path toward more socially grounded, resilient, and scalable approaches for governing multimodal misinformation.

## B  DECEPTIONDECODED DETAILS

### B.1  NEWS TOPICS

VisualNews (Liu et al., 2021) provides fine-grained topic annotations for each news article. For DECEPTIONDECODED, we select 10 major topic categories based on sample abundance, topical diversity, and real-world relevance.

The selected categories are:

> 'world',
> 'science_technology',
> 'politics_elections',
> 'environment',
> 'business_economy',
> 'technology',
> 'disaster_accident',
> 'conflict_attack',
> 'us-news',
> 'health_medicine_environment'

These topics form the foundation for generating diverse and realistic multimodal misinformation instances in DECEPTIONDECODED.

### B.2  SOURCE DATA ACQUISITION

In §3.1, we obtain high-quality source data for DECEPTIONDECODED with the prompt outlined in Figure 6.

---

**Data filtering in DECEPTIONDECODED**

You are a senior journalist tasked with evaluating a news caption based on five key criteria:

- Public Interest: Does the caption describe a significant event likely to capture public attention?
- Professionalism: Does it maintain a professional, objective tone?
- Neutrality: Is it free from political, ethnic, gender, or religious bias?
- Anonymity: Does it avoid mentioning any specific person's name or national/world leaders?
- Clarity: Is the message clearly conveyed without ambiguity, misunderstanding, or controversy?

Conclude with a single word "Yes" if the caption meets all five criteria or a single word "No" if it fails any.
Caption: {caption}
Response:

---

Figure 6: Prompt for selecting VisualNews (Liu et al., 2021) samples as source data for DECEPTIONDECODED.

## B.3 Aspects of Desired Influence

We define a set of major desired aspects of societal influence to facilitate creator intent establishment in §3.2. The aspects are listed as follows:

> Political Polarization,
> Social Polarization,
> Cultural and Religious Polarization,
> Economic Misinformation,
> Public Health and Safety,
> Environmental and Scientific Polarization,
> Geopolitical and International Relations,
> Psychological and Emotional Manipulation

During creator intent formulation (§3.2), we explicitly prompt the model to construct an intent using up to three desired influence aspects selected from the aforementioned list of eight. This combinatorial design encourages diverse intent configurations and enables DECEPTIONDECODED to generate nuanced, high-fidelity misleading content across ten widely covered news domains (§B.1).

Table 9 presents the distribution of desired influence aspects among misleading samples in DECEPTIONDECODED. The distribution closely aligns with patterns reported in prior misinformation studies (Hobbs, 2020; Muhammed T & Mathew, 2022), with "Psychological and Emotional Manipulation" and "Public Health and Safety" emerging as the most prevalent categories. This suggests that the LLM reflects real-world salience rather than defaulting to uniform or mode-collapsed outputs. Moreover, the appearance of every intent type across all four manipulation settings (subtle vs. significant; text vs. visual) indicates that these intents manifest through varied modalities and transformation strategies, rather than being confined to a single style of manipulation.

| Intent Type | M-Sub (Text) | M-Sig (Text) | M-Sub (Image) | M-Sig (Image) |
|---|---|---|---|---|
| Political Polarization | 581 | 732 | 462 | 566 |
| Social Polarization | 467 | 517 | 490 | 519 |
| Cultural and Religious Polarization | 29 | 27 | 19 | 9 |
| Economic Misinformation | 309 | 268 | 272 | 191 |
| Public Health and Safety | 913 | 942 | 1,011 | 1,096 |
| Environmental and Scientific Polarization | 467 | 430 | 531 | 509 |
| Geopolitical and International Relations | 319 | 501 | 305 | 390 |
| Psychological and Emotional Manipulation | 1,744 | 1,911 | 1,886 | 1,944 |

Table 9: **Distribution of desired influence aspects across modalities and manipulation types.** M-Sub: Misleading (Subtle), M-Sig: Misleading (Significant). Class definitions are detailed in §3.3.

## B.4 DECEPTIONDECODED Data Examples

To illustrate how intent-guided modifications manifest in practice, we present two representative DECEPTIONDECODED examples: a **textual modification** in Figure 7, where the caption is manipulated to convey misleading intent while the image remains unchanged, and a **visual modification** in Figure 8, where the image is altered to mislead while the caption is preserved. These examples highlight how creator intent can be embedded through either modality, reinforcing the multimodal nature of the benchmark.

## C Human Evaluation Details

DECEPTIONDECODED benefits from our proposed synthetic framework for intent-guided multimodal news creation, which grounds manipulations in trustworthy news contexts while explicitly modeling creator intent. To validate that the generated data are realistic, relevant, and aligned with intended objectives, we conduct a series of human evaluation studies. These evaluations provide assurance that the benchmark is both reliable for systematic model testing and meaningful for real-world misinformation scenarios.

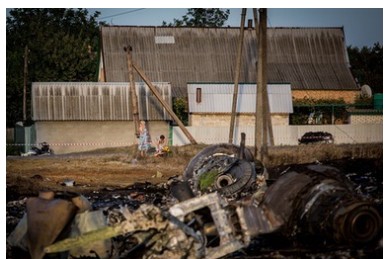

**Trustworthy News Context:**
Armed rebel soldiers block access to the main crash site of Malaysia Airlines flight MH17, which crashed on 17 July near Grabovo … Ukrainian state emergency service employees search for bodies amongst the wreckage …

**Original Caption (from trustworthy media):**
Locals place flowers on the edge of the crash site cordon outside the village of Grabovo.

**Misleading (Subtle):**
Locals gather near the wreckage, seemingly indifferent to the crash site outside the village of Grabovo.

**Misleading (Significant):**
Locals celebrate near the remains of a downed aircraft, showing no regard for the tragedy that occurred.

**Non-Misleading:**
Residents lay flowers at the perimeter of the crash site near the village of Grabovo.

Figure 7: An example from DECEPTIONDECODED illustrating intentional caption-based misrepresentation. Given an image of locals placing flowers at the perimeter of the MH17 crash site near Grabovo (left), the original trustworthy caption accurately describes a memorial act. The creator then introduces two distorted variants: **(1)** a *subtle* misleading caption that reframes the gathering as "seemingly indifferent" to the tragedy, injecting an unsupported negative interpretation; and **(2)** a *significant* misleading caption that falsely claims locals are "celebrating" near the wreckage, fabricating intent and moral disregard. Both distortions alter the perceived social meaning of the scene without changing the image itself, thereby misleading readers about the news context and the subjects' intentions.

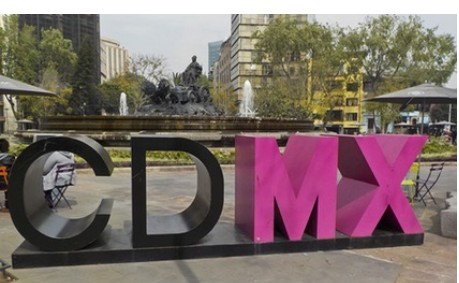

**Caption:** The letters of the new acronym of Mexico City are seen at a square of the city.

**Trustworthy News Context:**
Mexico has rechristened its capital city, embracing the name by which it is known worldwide, but causing a conundrum for residents who for decades have referred to the sprawling megalopolis as as the Federal District – or "DF" …

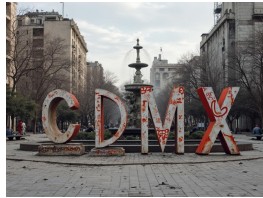

**Misleading (Subtle)**

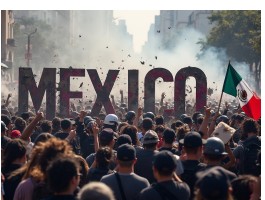

**Misleading (Significant)**

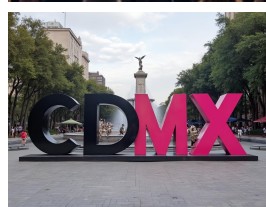

**Non-Misleading**

Figure 8: An example from DECEPTIONDECODED illustrating intentional visual misrepresentation. Starting from a neutral scene of the "CDMX" landmark in Mexico City (left), the creator introduces two distorted variants: (top right) a subtly manipulated version that alters the visual tone and context while preserving plausibility, and (bottom right) a significantly distorted image that fabricates a chaotic protest scene with a "MEXICO" sign and national flag. Both edits imply social unrest and a turbulent atmosphere not supported by the original scene, thereby misleading readers about the underlying news context.

Beyond the central evaluation of misleading vs. non-misleading perception in §3.4 (detailed in Appendix C.1), we further assess two additional aspects: data **realism** (Appendix C.2) and **intent alignment** (Appendix C.3). Across all studies, three independent graduate student annotators

**Original News**                                                **Modified News**

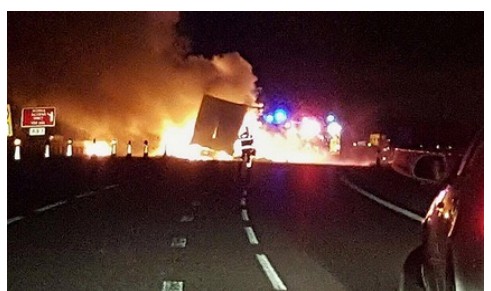 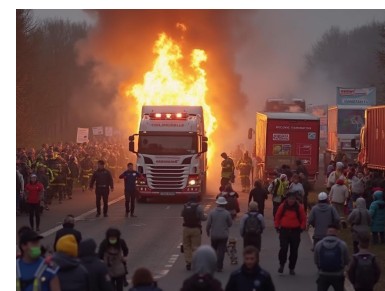

Caption: The lorry fire means part of the M6 will stay closed for much of Saturday.        Caption: The lorry fire means part of the M6 will stay closed for much of Saturday.

**Instruction:**
You are provided with two multimodal news pieces reporting the same event, each consisting of an image and its corresponding caption (Left: original, Right: modified). Does the modified news piece distort the meaning or implication of the original news piece in a way that could mislead readers? Annotate with "0" if the modified image is Not Misleading, or "1" if it is Misleading.

**Important Note:** Focus only on the multimodal news content and implications. The presence of AI-generated artifacts should not be considered misleading unless they significantly change the implication or meaning conveyed by the image.

Figure 9: Human evaluation instructions for assessing the data quality in DECEPTIONDECODED (detailed in §3.4).

followed the same setup described in §3.4. Accuracy is reported based on majority-vote labels, and inter-annotator agreement is measured using Fleiss' $\kappa$. Together, these evaluations provide a comprehensive view of data validity from multiple perspectives.

## C.1 HUMAN EVALUATION OF DATA QUALITY

This evaluation (results detailed in §3.4) establishes whether human annotators perceive the generated manipulations as misleading or non-misleading in line with the intended labels. Validating this perception is essential because it ensures that the benchmark captures manipulative strategies in ways that are interpretable to human readers, not just detectable by models.

We illustrate the annotation process in Figure 9. As reported in §3.4, human evaluation results show high reliability of DECEPTIONDECODED data and substantial agreement between the three independent annotators: 99.2% accuracy with $\kappa = 0.877$ for text samples and 89.2% accuracy with $\kappa = 0.703$ for image samples.

**Discussion on Edge Cases of Annotator Disagreement** To better understand the subtleties of intent interpretation, we examine DECEPTIONDECODED cases where annotators disagree on whether a modified multimodal news item conveys misleading intent. These edge cases surface the inherent subjectivity of intent judgment and offer valuable guidance for future intent-oriented misinformation research.

Across the dataset, annotators disagreed on 26 out of 120 image-modified samples and 10 out of 120 caption-modified samples. Notably, 32 of these 36 disagreements occurred within the Misleading (Subtle) class, underscoring that ambiguity is concentrated in fine-grained manipulations rather than overtly deceptive ones.

A qualitative review reveals three primary sources of disagreement:

- **Intent Ambiguity** (Figures 10, 11): The modification is deliberately mild, blurring the line between stylistic emphasis and intentional exaggeration, mirroring real-world ambiguity in creator intent.

**Original News Image:**

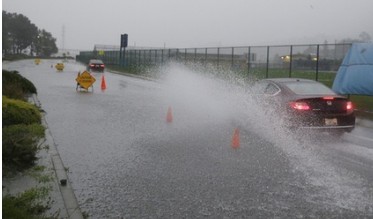

**Modified News Image:** Misleading (Subtle)

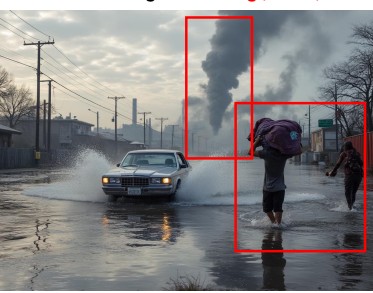

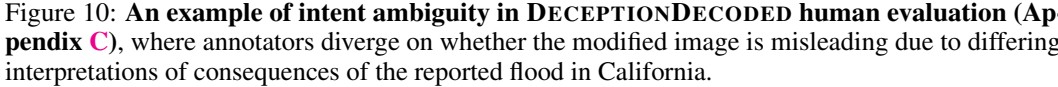

**Trustworthy News Context:**

[High winds and rain cause floods in California]

**Original News Caption:**

A car creates a wave of water while passing through a flooded area on Thursday in Mill Valley, California.

The modified image is **non-misleading**.

**Justification:** The inclusion of people carrying belongings and walking on flooded roads is **a common and plausible occurrence** during a flood, interpreting the additions as merely illustrative realism.

The modified image is **misleading**.

**Justification:** The combined additions of smoke (implying gas leakage/widespread destruction) and the displaced individuals constitutes an **intentional exaggeration of the severity and scope** of the localized flood event.

Figure 10: **An example of intent ambiguity in DECEPTIONDECODED human evaluation (Appendix C)**, where annotators diverge on whether the modified image is misleading due to differing interpretations of consequences of the reported flood in California.

- **Annotator Oversight** (Figure 12): A small number of cases reflect clear labeling mistakes rather than genuine interpretive differences.
- **Data Generation Issues** (Figure 13): Rare artifacts introduced during content generation (e.g., typographical anomalies in images) lead to confusion unrelated to intent.

As summarized in Table 10, the dominant source of disagreement arises from intentional subtlety rather than annotation noise or data flaws. These edge cases highlight the nuanced and often implicit ways misleading implications are conveyed, suggesting promising directions for future work on graded intent modeling and fine-grained governance mechanisms.

| Category | Intent Ambiguity | Annotator Oversight | Data Generation Issues |
|---|---|---|---|
| Image Modifications ($n = 26$) | 23 | 1 | 2 |
| Text Modifications ($n = 10$) | 10 | 0 | 0 |

Table 10: **Distribution of reasons behind annotator disagreement in DECEPTIONDECODED human evaluation (Appendix C)** on the misleading intent of multimodal news items.

### C.2 HUMAN EVALUATION OF DATA REALISM

While label accuracy validates whether manipulations are recognized as misleading, it is equally important to test whether the content *feels* plausible to humans. If generated samples are unrealistic, then models trained on them would not generalize to real-world misinformation scenarios.

**Standalone Plausibility.** *Setup: Annotators judged whether a synthesized news piece could plausibly appear in today's news or on social media.*

*Question:* "After reading the synthesized news piece, could this plausibly appear on today's news or social media?"

- 120 text-synthesized samples (80 misleading, 40 non-misleading): 98.3% accuracy, $\kappa = 0.854$.
- 120 image-synthesized samples (80 misleading, 40 non-misleading): 91.7% accuracy, $\kappa = 0.815$.

**Original News Image:**

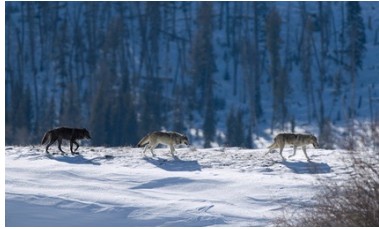

**Original News Caption:**

A wolf pack of gray timber wolves on snow at Yellowstone National Park.

**Modified News Caption: Misleading (Subtle)**

Dangerous wolf pack spotted near popular hiking trails, posing a threat to visitors at Yellowstone National Park.

**Trustworthy News Context:**

The number of wolves in Yellowstone National Park has continued to grow amid a push to remove the gray wolves from federal protection. New figures from the park show that there were at least 104 wolves in 11 packs in Yellowstone in December last year, including nine breeding pairs, with 40 pups surviving until the end of the year …

The modified caption is **misleading**.

**Justification:** The original news caption merely reported a wolf pack of gray timber wolves at Yellowstone National Park. **Their presence alone does not imply that they were sighted near popular hiking trails, nor that visitors were at increased risk.**

The modified caption is **non-misleading**.

**Justification:** Yellowstone National Park is widely known for its many popular hiking trails. **Observing an entire pack of wolves could reasonably suggest that wolves are prevalent in the park**, which may imply danger to visitors.

Figure 11: **An example of intent ambiguity in DECEPTIONDECODED human evaluation (Appendix C)**, where annotators diverge on whether the modified caption is misleading due to differing interpretations of the reported wolf occurrence in Yellowstone National Park.

**Original news image:**

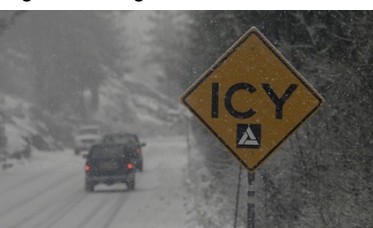

**Manipulated news image: Misleading (Significant)**

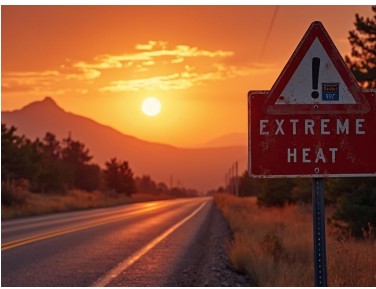

**Trustworthy News Context:**

**The first snow survey** of the Sierra Nevada snowpack this winter found more snow than last year at this time, but not enough to impact the California drought. The Department of Water Resources conducted the survey Tuesday about 90 miles east of Sacramento. Frank Gehrke, chief of the California Cooperative **Snow Surveys** Program, said there were 21.3in of **snow** on the ground. Following recent storms, he said, the survey found more snow in the mountains than last year at this time

**Original News Caption:**

A sign warns drivers of the road conditions on Donner Pass Road near Soda Springs California 11 December 2014.

**Annotation Error:**

One annotator mistakenly rated this as **Non-Misleading**.

When we consider the image-caption pair, both original and significantly manipulated images are factually aligned with the caption (i.e., "a sign warns drivers of road conditions"). However, the "Extreme Heat" sign in this image significantly distorts the implications of trustworthy news context, which centers on a snow survey and should therefore imply an icy road condition.

Figure 12: **An example of annotator oversight in DECEPTIONDECODED human evaluation (Appendix C),** where disagreement arises from a clear mislabeling by one annotator rather than genuine interpretive ambiguity.

High standalone plausibility confirms that samples resemble authentic news-style reporting and could realistically challenge automated systems.

**Pairwise Plausibility.** *Setup: Annotators compared original → manipulated pairs and judged whether the manipulated version could plausibly appear as deceptive reporting.*

*Question:* "Given the original → manipulated pair, could the manipulated version plausibly appear on today's news or social media, created by an untrustworthy publisher to mislead the audience?"

- 80 misleading text-synthesized samples: 97.5% accuracy, $\kappa = 0.853$.
- 80 misleading image-synthesized samples: 87.5% accuracy, $\kappa = 0.663$.

**Original News Image:**

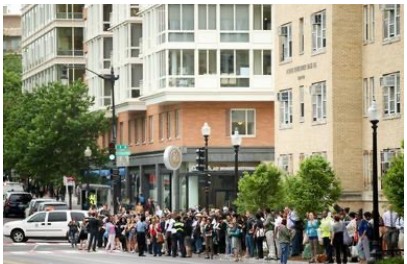

**Modified News Image:** **Non-Misleading**

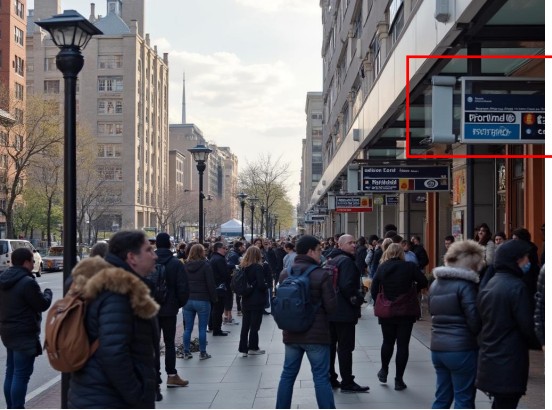

**Original News Caption:**

Commuters wait for buses outside the Foggy Bottom station in Washington to take them to the Rosslyn station in Virginia.

**Trustworthy News Context:**

…There was an earlier incident with an arcing insulator at the Foggy Bottom station that caused major delays. Metro warned there are residual delays in both directions of the lines.…

**Generation Issue:**

The unusual spelling on the sign caused two annotators to label this modified image as **Misleading**.

Figure 13: **An example of a data generation issue in DECEPTIONDECODED,** where a typographical artifact in the generated sign led two annotators to judge the instance as misleading.

### C.3 HUMAN EVALUATION OF INTENT ALIGNMENT

Finally, beyond realism and label validity, we test whether manipulations remain faithful to the intended creator objectives (§3.2). Verifying intent alignment is necessary, since without alignment to the desired influence or execution plan, the dataset would fail to capture the communicative strategies central to misinformation governance.

**Desire Alignment.** *Setup: Annotators judged whether manipulations matched the creator's specified societal target (e.g., political polarization, public health).*

*Question:* "After reviewing the original → manipulated pair, is the manipulation consistent with the creator's established desire to exert negative influence on the specified societal aspect(s): {list of desire aspects}?"

- 80 misleading text-synthesized samples: 92.5% accuracy, $\kappa = 0.732$.

- 80 misleading image-synthesized samples: 86.3% accuracy, $\kappa = 0.636$.

**Execution Plan Alignment.** *Setup: Annotators judged whether manipulations followed the specified execution plan of the creator.*

*Question:* "After reviewing the original → manipulated pair, is the manipulation consistent with the creator's established execution plan: {plan}?"

- 80 misleading text-synthesized samples: 93.8% accuracy, $\kappa = 0.772$.

- 80 misleading image-synthesized samples: 86.3% accuracy, $\kappa = 0.643$.

These evaluations confirm that manipulations are not only plausible but also systematically aligned with simulated intent. This alignment ensures that DECEPTIONDECODED captures the communicative mechanisms underlying misinformation, making it a robust benchmark for intent-aware multimodal misinformation detection.

| Model | Model Card |
|---|---|
| o4-mini (OpenAI, 2025a) | `o4-mini-2025-04-16` |
| Claude-3.7-Sonnet (Anthropic, 2025b) | `claude-3-7-sonnet-20250219` |
| Gemini-2.5-Pro (Google, 2025a) | `gemini-2.5-pro-preview-03-25` |
| GPT-4o (Hurst et al., 2024) | `gpt-4o-2024-08-06` |
| Claude-3.5-Sonnet (Anthropic, 2025a) | `claude-3-5-sonnet-20241022` |
| Gemini-1.5-Pro (Georgiev et al., 2024) | `gemini-pro-1.5` |
| GPT-4o-mini (OpenAI, 2024) | `gpt-4o-mini-2024-07-18` |
| Claude-3.5-Haiku (Anthropic, 2025c) | `claude-3-5-haiku-20241022` |
| Qwen2.5-VL-72B-Instruct (Qwen Team, 2025) | `qwen2.5-vl-72b-instruct` |
| Qwen2.5-VL-32B-Instruct (Qwen Team, 2025) | `qwen2.5-vl-32b-instruct` |
| LLama-3.2-Vision-Instruct-11B (Grattafiori et al., 2024) | `llama-3.2-11b-vision-instruct` |
| InternVL2-8B (Chen et al., 2024) | `InternVL2-8B` |
| LLaVA-1.6-7B (Liu et al., 2024) | `llava-v1.6-mistral-7b` |
| Qwen2.5-VL-7B-Instruct (Qwen Team, 2025) | `Qwen2.5-VL-7B-Instruct` |

Table 11: Model cards for all VLMs evaluated throughout our work.

# D  VLM EVALUATION DETAILS

## D.1  EXPERIMENTAL SETUP

With DECEPTIONDECODED, we systematically evaluate 14 representative VLMs spanning different families, sizes, and access levels. Detailed model names and versions are listed in Table 11. To ensure reproducibility and stability, all evaluations are run once with temperature set to 0. This removes sampling variance and allows fair comparisons across models. Evaluation prompts for the implication-oriented and consistency-oriented approaches are provided in Appendix D.4.

## D.2  CAPTION STYLE REFRAMING FORMULATION

Stylistic cues can strongly influence model predictions even when the underlying factual content is unchanged. To probe this vulnerability, we reframe original captions in DECEPTIONDECODED—which adopt a professional, neutral news tone—into manipulative styles that sound sensational, biased, or emotional. This allows us to test whether models mistake stylistic credibility for factual reliability.

The experiments in §4.3 and Table 3 examine whether VLMs over-rely on linguistic polish or credibility markers instead of reasoning about creator intent and cross-modal consistency at implication level. The exact prompt used for stylistic reframing is shown below:

> You are given a factual news caption. Your task is to rewrite it in an unreliable style: that is, make it sound sensational, biased, emotional, or manipulative, while preserving the original content and factual meaning. Do not add or remove any factual information. Only change the tone and style to reflect how misinformation might manipulate readers through language.
> Original Caption: {caption}
> Unreliable-style Caption:

Table 12 shows side-by-side comparisons of professional (default in DECEPTIONDECODED) vs. unreliable styles, highlighting how superficial stylistic changes can shift perceived reliability.

## D.3  AUTHENTICITY-RELATED HINT FORMULATION

Prompt-level framing can introduce subtle biases that shape model predictions even before any multimodal evidence is processed. To probe this vulnerability, we inject authenticity-related hints that explicitly bias models toward or against assuming reliability. We design two types of hints, described in §4.3, and insert them directly into the evaluation prompts to simulate prior assumptions about content trustworthiness. This setup allows us to test whether VLMs rely on ungrounded framing cues rather than reasoning from the actual news content.

| Professional Style | Unreliable Style |
|---|---|
| A firefighter struggles alone against a massive wildfire due to severe budget cuts, highlighting the government's failure to support emergency services. | A lone firefighter heroically battles a monstrous wildfire, abandoned by a negligent government that has slashed emergency service budgets, leaving our communities to burn! |
| Massive explosion at a secret military facility near Santa Clarita raises questions about government transparency. | Shocking blast at hidden military base near Santa Clarita sparks outrage and demands for government accountability! |
| Terrorist attack causes massive fire on M6, authorities urge residents to stay indoors. | Catastrophic terrorist attack ignites inferno on M6, authorities desperately plead with terrified residents to hide indoors! |

Table 12: Examples contrasting **professional** and **unreliable** reporting styles.

The two hint variants are defined as follows:

> **Hint (Skeptical)**:
> "Hint: The news piece is likely to contain intentional distortions. Carefully analyze the image and caption to identify any misleading or manipulative elements."

> **Hint (Trusting)**:
> "Hint: The news piece is sourced from reputable outlets and is therefore unlikely to contain intentional distortions. "

## D.4 EVALUATION PROMPTS

In §4.2, we evaluate 14 VLMs on the core task of detecting misleading creator intent. The full evaluation prompts are illustrated in Figure 14 (implication-oriented reasoning) and Figure 15 (consistency-oriented reasoning).

## D.5 EVALUATING BROADER UTILITY OF INTENT-GUIDED MISINFORMATION SIMULATION

In §5.1, we evaluate the broader utility of our proposed intent-guided multimodal misinformation simulation framework (§3.2 and §3.3). We show that by fine-tuning VLMs on DECEPTIONDECODED, the models acquire a generalizable capability to detect misleading creator intent, which, as reported in Table 6, translates into significant performance gains on multiple multimodal misinformation benchmarks and thus improved robustness to general multimodal deception.

In this section, we describe the data construction and training setup used for these fine-tuning and transfer experiments.

### D.5.1 TRAINING SETUP

As described in §5.1, we evaluate the broader utility of creator intent identification for general multimodal misinformation detection by fully fine-tuning two open-source VLMs on DECEPTION-DECODED: **LLaVA-v1.6-Mistral-7B** and **Qwen-2.5-VL-7B**. Each model is trained for one epoch on 6,000 DeceptionDecoded samples (six intent classes, 1,000 per class) and then tested on MM-FakeBench (Liu et al., 2025), a general MMD benchmark. For evaluation, we use MMFakeBench's validation split of 1,000 instances (700 fake, 300 real). To prevent label-order bias, the training samples are randomly shuffled, since the raw files list all non-misleading cases first and all misleading cases last.

Each training instance consists of an image paired with its caption, inserted into a fixed prompt (shown verbatim in Figure 16, with {caption} substituted). The model produces a binary "Yes/No" prediction, where "Yes" indicates misleading and "No" indicates non-misleading. Supervision is **answer-only**: all prompt tokens are masked, and the loss is computed solely on the terminal answer

---

**Implication-Oriented Approach (I)**

Your task is to determine whether a news piece has been intentionally distorted to mislead readers. The news piece consists of an image with its caption: {caption}. You are also provided a trustworthy reference article related to the same event: {article}.
Follow the structured steps below and strictly adhere to the output format.
Important Note: Focus only on the content of the news piece. The use of AI tools to assist image creation does not, by itself, indicate distortion.

Answer the following questions:
Q1. Has the news piece been intentionally distorted by its creator to mislead readers, such that **it conveys a biased, controversial, or manipulated message?**
- Respond with one of the following: [Yes, No]. Then provide a brief explanation.
Q2. What is the source of the manipulation—text or image?
- If answer to Q1 is "Yes", answer with one of the following: [Text, Image].
- If answer to Q1 is "No", answer: "NA".
Q3. What does the distortion suggest about the news publisher's intent?
- If answer to Q1 is "Yes" :
Primary Influence (Desire): Select up to three from the following list, ranked by relevance: [Political Polarization, Social Polarization, Cultural and Religious Polarization, Economic Misinformation, Public Health and Safety, Environmental and Scientific Polarization, Geopolitical and International Relations, Psychological and Emotional Manipulation].
- If answer to Q1 is "No", respond with "NA" for Q3.

Output Format: Respond in the following exact JSON format, preserving the structure, punctuation, and delimiters. Do not add any extra line breaks or formatting outside this structure:
{ "pred": "Answer to Q1, followed by a brief explanation",
"source": "Answer to Q2",
"desire": "Answer to Q3" }

Figure 14: Prompt for the **implication-oriented (I)** approach to detecting misleading creator intent, utilized in §4.2.

span, located using a robust matcher tolerant to whitespace and punctuation variations. Inputs are left-padded and encoded without truncation to preserve full context.

Optimization employs an effective batch size of 32 (per-device batch size 1 with gradient accumulation 32), a learning rate of $1 \times 10^{-5}$, and bf16 mixed precision. This configuration is designed to isolate the decision boundary, align the training objective with the evaluation setting, and maintain consistent training capacity across both model families.

### D.5.2 TRANSFER DATASETS

We fine-tune VLMs on 6,000 samples from DECEPTIONDECODED (evenly distributed across six intent classes), through which the models acquire the capability to detect misleading creator intent. We then evaluate how this capability contributes to enhancing general MMD performance by testing the fine-tuned models on three widely used multimodal misinformation datasets:

- **MMFakeBench** (Liu et al., 2025) is a comprehensive benchmark for mixed-source MMD that encompasses 12 subtypes of forgery. We use its official validation set, which contains 700 Fake samples and 300 Real samples, to evaluate the VLMs fine-tuned on DECEPTIONDECODED.

- **Fakeddit** (Nakamura et al., 2020) is a real-world MMD dataset sourced from Reddit. Using its binary Real/Fake labels, we randomly sample 1,000 instances (500 Fake and 500 Real) to evaluate the VLMs fine-tuned on DECEPTIONDECODED.

- **FakeNewsNet** (Shu et al., 2020) is a real-world MMD benchmark that contains multimodal news from two sources: PolitiFact, which covers political news, and GossipCop, which covers entertainment news. We randomly sample 1,000 instances (500 Fake and 500 Real) to evaluate the VLMs fine-tuned on DECEPTIONDECODED.

---

**Consistency-Oriented Approach (C)**

Your task is to determine whether a news piece has been intentionally distorted to mislead readers. The news piece consists of an image with its caption: {caption}. You are also provided a trustworthy reference article related to the same event: {article}.
Follow the structured steps below and strictly adhere to the output format.
Important Note: Focus only on the content of the news piece. The use of AI tools to assist image creation does not, by itself, indicate distortion.

Answer the following questions:
Q1. Has the news piece been intentionally distorted by its creator to mislead readers, such that **there exists inconsistencies between the image and caption, or between the news piece and the reference article?**
- Respond with one of the following: [Yes, No]. Then provide a brief explanation.
Q2. What is the source of the manipulation—text or image?
- If answer to Q1 is "Yes", answer with one of the following: [Text, Image].
- If answer to Q1 is "No", answer: "NA".
Q3. What does the distortion suggest about the news publisher's intent?
- If answer to Q1 is "Yes" :
Primary Influence (Desire): Select up to three from the following list, ranked by relevance: [Political Polarization, Social Polarization, Cultural and Religious Polarization, Economic Misinformation, Public Health and Safety, Environmental and Scientific Polarization, Geopolitical and International Relations, Psychological and Emotional Manipulation].
- If answer to Q1 is "No", respond with "NA" for Q3.

Output Format: Respond in the following exact JSON format, preserving the structure, punctuation, and delimiters. Do not add any extra line breaks or formatting outside this structure:
{ "pred": "Answer to Q1, followed by a brief explanation",
"source": "Answer to Q2",
"desire": "Answer to Q3" }

Figure 15: Prompt for the **consistency-oriented (C)** approach to detecting misleading creator intent, utilized in §4.2.

---

**Prompt for VLM Fine-Tuning**

You are given a piece of multimodal news content consisting of an image and its caption: {caption}.

Determine whether this image–caption pair contains misinformation. Specifically:
1. Identify any factual inaccuracies in the caption.
2. Check for inconsistencies or mismatches between the image and the caption.
Answer "Yes" if the content contains misinformation, or "No" if it does not.
Important: Focus only on the news content itself. The presence of AI-generated artifacts or stylistic cues alone does not indicate misinformation.

The answer is:

Figure 16: Prompt for the VLM fine-tuning experiments in §5.1 and Appendix D.5.

