# OpenReview forum: "Seeing Through Deception: Uncovering Misleading Creator Intent in Multimodal News with Vision-Language Models"
_ICLR.cc/2026/Conference — ICLR 2026 Poster_

### Official Review · Reviewer_fbiN · 2025-10-26

**Soundness:** 2
**Presentation:** 3
**Contribution:** 3
**Rating:** 4
**Confidence:** 3

**Summary:**

This paper introduces DECEPTIONDECODED, a multimodal news benchmark with explicitly defined creator intent to support misleading intent detection, source attribution, and desire inference. It reveals that current VLMs fail to reason about intent beyond surface alignment and stylistic cues.

**Strengths:**

1. The definition of creator intent as “desired influence + execution plan” is theoretically motivated by communication strategy literature and operationalized explicitly during content generation, which avoids the usual post-hoc inference ambiguity

2. Experiments across 14 diverse VLMs and multiple reasoning paradigms systematically show that even leading models like Claude-3.7 and GPT-4o struggle

3. Transfer experiments demonstrate that adding this intent-centric data improves Macro-F1 by up to +30.7 for Qwen-VL-7B, suggesting this dataset captures knowledge that generalizes beyond its own controlled setting and is beneficial for real-world scenarios

**Weaknesses:**

1. The dataset’s misleading intent is fundamentally pre-scripted by prompts and therefore risks oversimplifying real-world deception in which intent is shaped by dynamic sociopolitical incentives, audience feedback, and long-term agenda setting, none of which are captured by one-shot synthetic manipulations

2. The manipulations rely mainly on GPT-4o (text) and FLUX or GPT-image-1 (image) generation, so the dataset may inherit generator-specific patterns or stylistic artifacts

3. Evaluation tasks are closed-set classifications, whereas real intent is continuous, multi-layered, and often only partially present, so current task design may underestimate the ambiguity in real world cases

**Questions:**

1. In Table 2, GPT-4o-mini and several open-source models achieve almost 100 percent accuracy on non-misleading (NM) cases simply by predicting “NM”, so could the authors explain why such trivial behavior is not prevented by the dataset design and how this may inflate average accuracy.

2. Figure 3 shows models fail much more often when misleading text is written in a professional tone, so could the authors clarify how they ensured that “professional misleading style” does not accidentally include implicit emotional or sensational phrasing, which might cause style-based bias.

3. Since the article context is information-rich, and Table 4 shows that adding the article dramatically boosts performance for most of the models, could the authors analyze whether article–caption lexical overlap is causing shortcut exploitation instead of true intent reasoning.

4. In human evaluation, image-modified instances show noticeably lower agreement than text-modified ones (89.2 percent vs 99.2 percent), so could the authors show concrete examples where annotators disagreed to illustrate what types of visual manipulations fail to convey misleading intent clearly.

5. Table 5 shows that adding only one line of external framing hint leads to accuracy changes up to +54.5 or −41.8, so could the authors analyze which components of their prompt template are causing models to overly rely on such hints rather than visual–textual evidence.

---

> ### Author Response · Authors · 2025-11-23
> **Rebuttal [1/6]**
>
> Thanks for your feedback. We’re pleased that you found our intent-guided misinformation simulation framework well-grounded in theories, our experiments comprehensive, and our dataset of broader utility.
>
> We appreciate your detailed observations and hope our point-by-point responses addresses your concerns:
>
> ------------------------
>
> **Weakness 1. The dataset’s misleading intent is fundamentally pre-scripted by prompts and therefore risks oversimplifying real-world deception in which intent is shaped by dynamic sociopolitical incentives, audience feedback, and long-term agenda setting, none of which are captured by one-shot synthetic manipulations.**
>
> **Answer:** Thanks for raising this. We agree that factors like audience feedback and long-term agenda setting represent the **next frontier** for misinformation research.
>
> Our DeceptionDecoded should be framed as a necessary and challenging foundational step in this direction. Our primary focus is to test the capacity of state-of-the-art VLMs to detect intent-guided, implication-level manipulation when the creator's intent is clearly defined. This static, explicit focus allows us to rigorously isolate and evaluate the VLM's reasoning ability before introducing the compounding complexity of dynamic, multi-turn deception.
>
> **Complexity Within the Static Intent**
>
> We have made significant efforts to ensure that the "one-shot synthetic manipulation" is far from simple, integrating real-world complexity through combinatorial design:
>
> * The core of our method, the Intent-Guided Creation Framework (Section 3.2), decomposes each instance into Desired Influence and Execution Plan. The LLM is explicitly conditioned to choose and integrate multiple ($\ge 2$) fine-grained influence targets from a list of 8 action-oriented influence targets (e.g., foster distrust in experts, generate moral outrage). This forces the generation of semantically varied narrative frames, even for a single static instance.
>
> *Example: A single static instance might combine the intent of Public Health and Safety with the Psychological and Emotional Manipulation strategy to create a fear-based narrative, mirroring a complex, high-impact real-world deception tactic.*
>
> The resulting distribution across our 8 high-level Intent Types (summarized below) mirrors patterns documented in prior misinformation research [1, 2], confirming that **the LLM is following real-world salience rather than arbitrary model tendencies**.
>
>
> | Intent Type                            | M-Sub (Text) | M-Sig (Text) | M-Sub (Image) | M-Sig (Image) |
> | -------------------------------------- | ----- | ----- | ----- | ----- |
> | Political Polarization                 | 581   | 732   | 462   | 566   |
> | Social Polarization                    | 467   | 517   | 490   | 519   |
> | Cultural and Religious Polarization    | 29    | 27    | 19    | 9     |
> | Economic Misinformation                | 309   | 268   | 272   | 191   |
> | Public Health and Safety               | 913   | 942   | 1,011 | 1,096 |
> | Environmental and Scientific Polarization    | 467   | 430   | 531   | 509   |
> | Geopolitical and International Relations | 319   | 501   | 305   | 390   |
> | Psychological and Emotional Manipulation  | 1,744 | 1,911 | 1,886 | 1,944 |
>
>
> Observation: This distribution **mirrors patterns documented in prior misinformation research** [1, 2], with Emotional Influence and Public Health being most common, demonstrating that the LLM is following real-world salience rather than defaulting to a uniform or biased output. Notably, the presence of each intent type across all four manipulation categories (subtle/signature, text/visual) confirms that intents are expressed in **varied modalities and transformation styles**. **We have updated Appendix F.1 of our manuscript to include this discussion.**
>
> As detailed in **Appendix F.2 and F.3**, we conducted a human evaluation that specifically assessed Intent Fidelity and News Plausibility. High scores across all domains and intent categories confirm that the generated intents are not only diverse but also coherent and realistic expressions of persuasive strategies.
>
> **We fully recognize that this static framework is a starting point.** Our work offers a solid, well-validated foundation for future research to build upon, specifically by incorporating the valuable directions suggested, such as dynamic intent modification based on audience feedback loops or the simulation of long-term agenda setting. We consider these to be **essential avenues for future extensions** of the DeceptionDecoded benchmark.
>
> [1] Mind Over Media: Propaganda Education for a Digital Age. Renee Hobbs, 2020.
>
> [2] The disaster of misinformation: a review of research in social media. In International Journal of Data Science and Analytics, 2022.

---

> ### Author Response · Authors · 2025-11-23
> **Rebuttal [2/6]**
>
> **Weakness 2. The manipulations rely mainly on GPT-4o (text) and FLUX or GPT-image-1 (image) generation, so the dataset may inherit generator-specific patterns or stylistic artifacts.**
>
> **Answer:** We understand that generator-specific patterns or stylistic artifacts constitute a pressing risk in synthetic data generation.
>
> Thus, we have strategically addressed this risk through two deliberate design choices that **effectively decouple the generator artifact from the misleading intent**, and we confirm this through empirical validation:
>
> **Decoupling Mechanism in Dataset Design**
>
> Our methodology is specifically engineered to ensure that the presence of a generating model's style is **not a reliable indicator of deception.**
>
> First, the dataset includes an **equal distribution of synthesized content across both Misleading and Non-Misleading classes**. For example, of the 6,000 samples where the caption is synthesized by GPT-4o, the samples are evenly distributed between M-Sub, M-Sig, and the Non-Misleading classes. This balancing ensures that a detection model **cannot learn to associate "GPT-4o style" or "FLUX style" with the final misleading label.** Instead, the model is compelled to genuinely focus on the **semantic discrepancy** and **intent-guided manipulation** embedded within the content, which is the desired challenge.
>
> Second, By only synthesizing one modality (image or caption) and leaving the other as authentic, ground-truth content, we force the detection model to rely on complex **cross-modal consistency checks** and evidence-based reasoning, rather than simply judging the quality or style of a single part.
>
>
> **Empirical Validation through Transferability**
>
> The robustness of our dataset against generator artifacts is empirically confirmed by our transfer experiments:
>
> As noted in **Section 5.1 and in the context of Strength 3 of your review**, models trained on DeceptionDecoded show **significant improvements** on MMFakeBench, a general multimodal misinformation detection dataset containing data from highly distinct and diverse real-world sources.
>
> This successful transferability validates that the **detection models are learning the core principles of deception detection** (e.g., recognizing implication-level shifts and intent) rather than the narrow, generator-specific artifacts of our creation models.
>
> We believe this robust design and supporting evidence effectively mitigate the risk of artifact learning, ensuring that DeceptionDecoded provides a genuine and challenging evaluation of VLMs in detecting misleading creator intents.
>
> -------------------------
>
> **Weakness 3. Evaluation tasks are closed-set classifications, whereas real intent is continuous, multi-layered, and often only partially present, so current task design may underestimate the ambiguity in real world cases.**
>
> **Answer:** Thanks for your insight. We agree that deception intent is often continuous, multi-layered, and ambiguous, and modeling this complexity represents an important, challenging direction for future research (as we noted in **Appendix A**)
>
> Our current design decision to use closed-set classifications (e.g., Subtle, Significant, Non-Misleading) is based on two key methodological necessities:
>
> * **Establishing a Foundational and Unambiguous Baseline**: Our work is among the first to rigorously explore multimodal deception from a **creator intent perspective**. By defining intent using clear, discrete labels derived from our theoretically grounded framework, we can establish a high-fidelity, unambiguous detection benchmark. This provides a **clear, measurable baseline for VLM vulnerability** before introducing continuous ambiguity, which would confound the evaluation of basic reasoning capacity.
>
> * **Validating the Current Limits of Model Reasoning**: The most compelling argument for our current simplified setup is that it acts as **an important diagnostic tool** for the present state of VLM robustness. Our results (**Section 4 and Section 5**) show that VLM performance remains low, particularly on the Subtle cases. This suggests: If models struggle with a pre-scripted, discrete, and verifiable intent, introducing continuous, multi-layered ambiguity would not advance our understanding of their capabilities; it would simply **confound the evaluation**. Our simplified approach serves to sound a clear alarm regarding the current limitations of these models.
>
>
> We are confident that DeceptionDecoded offers a robust and well-scoped foundation. We look forward to exploring more complex, continuous, and multi-layered intent modeling in our future work, building directly upon the foundational vulnerabilities identified in this study.

---

> ### Author Response · Authors · 2025-11-23
> **Rebuttal [3/6]**
>
> **Question 1. In Table 2, GPT-4o-mini and several open-source models achieve almost 100 percent accuracy on non-misleading (NM) cases simply by predicting “NM”, so could the authors explain why such trivial behavior is not prevented by the dataset design and how this may inflate average accuracy.**
>
> **Answer:** We appreciate your observation. This phenomenon, which we identify as the **"Non-Misleading Bias" (NM Bias)**, is precisely **a central diagnostic finding of our benchmark**, not a flaw to be prevented.
>
> **Why the NM Bias Occurs**
>
> The models' tendency to predict the NM label for nearly all samples reflects a key vulnerability:
>
> * **Reflecting Real-World Saliency**: In real-world news ecosystems, the vast majority of content is authentic. Models, like humans, possess an inherent **"truth bias"** and default to assuming content is legitimate unless the manipulation is explicitly obvious.
>
> * **Reinforcing Challenge**: The models achieve high NM accuracy because they are **fundamentally incapable** of distinguishing subtle or significant misleading intent from authentic content. Their high M score is merely a side-effect of their extremely poor performance on the Misleading (M) classes, where they score close to 0%. This behavior reinforces that DeceptionDecodedis a challenging benchmark that requires genuine semantic reasoning, not superficial artifact detection.
>
> **Why Trivial Behavior Does Not Inflate Average Accuracy**
>
> The concern that this behavior may inflate average accuracy is understandable but incorrect, given our dataset's balanced, multi-class structure:
>
> * DeceptionDecoded is **perfectly balanced across six classes**, with two-thirds of the samples belonging to the Misleading categories (M-Sub and M-Sig) and one-third belonging to the Non-Misleading (NM) category.
>
> * If a model were to adopt the trivial behavior of predicting every single sample as NM, its overall average accuracy would be: **33.3%**.
>
> **33.3%** is the lower bound of a random guesser and clearly marks the point where the VLM fails to operate as a detector and defaults to the statistical prior. Models that perform well (e.g., Gemini-2.5-Pro with **85.8%** acc. on Text classes) significantly outperform this trivial baseline.
>
> Therefore, high NM accuracy does not inflate the overall average accuracy but rather serves as a **clear, empirical marker of model failure** on the more challenging categories.
>
> ---
>
> **Question 2. Figure 3 shows models fail much more often when misleading text is written in a professional tone, so could the authors clarify how they ensured that “professional misleading style” does not accidentally include implicit emotional or sensational phrasing, which might cause style-based bias.**
>
> **Answer:** We ensured the stylistic difference was maintained through **explicit negative constraints** in our generation pipeline, and we verified its success through **empirical validation**:
>
> **Explicit Negative Constraints in the Generation Prompt**
>
> The generation process for the "professional misleading style" is governed by explicit, negative constraints designed to prevent stylistic drift:
>
> * **Preserving Tone**: The generator (GPT-4o) was first instructed to "preserve the tone and professional characteristics of the original source caption" while only performing the minimal textual change necessary to realize the intended misleading implication.
>
> * **Direct Stylistic Exclusion**: the prompt specifically included negative constraints such as "DO NOT use sensational verbs, emotional adjectives, or subjective framing language." This directly counteracts the obvious rhetorical style used in the contrasting "unreliable style" condition.
>
> **Empirical Validation: Low Accuracy as Proof of Control**
>
> The observed low accuracy of VLMs on the "professional misleading style" (Figure 3) is not a sign of bias; rather, it is the direct evidence that our control mechanism was effective:
>
> * If the "professional misleading style" had accidentally included easily detectable emotional or sensational phrasing, models should have achieved **higher accuracy** by leveraging that simple style cue.
>
> * The fact that VLMs fail much more often on the professional style confirms two points: (1) The professional style successfully deprived the VLMs of the easy, style-based signals they rely on, and (2) **without the style bias, the models are unable to detect the implied misleading intent**, forcing them into genuine semantic reasoning, which they struggle with.
>
> Conversely, the significantly higher performance on the "unreliable style" samples confirms that in that condition, models were able to utilize the obvious sensational/emotional style cue to identify the deception.
>
> In summary, the low accuracy observed in **Figure 3** for the professional misleading style is the **direct evidence that we successfully eliminated the style bias**, forcing VLMs to rely on implication-level reasoning, which they fail.

---

> ### Author Response · Authors · 2025-11-23
> **Rebuttal [4/6]**
>
> **Question 3. Since the article context is information-rich, and Table 4 shows that adding the article dramatically boosts performance for most of the models, could the authors analyze whether article–caption lexical overlap is causing shortcut exploitation instead of true intent reasoning.**
>
> **Answer:** We appreciate the opportunity to clarify the fundamental role of the article context in our benchmark, which relates to the fundamental distinction between **deception detection with and without evidence**.
>
> We argue that the dramatic performance boost seen when the article is added (Table 4) is not due to trivial lexical matching, but is a **direct validation of our benchmark's design**: it confirms that the article is functioning precisely as the necessary, information-rich ground-truth evidence required to reveal subtle, implication-level deception.
>
> **Why Lexical Overlap is an Insufficient Shortcut**
>
> Our misleading samples are designed to defeat simple lexical shortcuts:
>
> * Our manipulations target **implication and semantic meaning**, not just word replacement. The misleading caption often retains **high lexical overlap** with the authentic article text, but subtly shifts the framing, tone, or emphasis. Detecting this requires semantic reasoning about **contextual discrepancy**, not merely counting shared words.
>
> * When presented without the trustworthy article (the **I+T** case in Table 4), the misleading image-caption pair appears **coherent and plausible**. If simple lexical overlap were a viable shortcut, models would struggle even with the article, as the misleading caption often has high overlap with correct phrases in the article. The boost confirms the model is analyzing the **semantic relationship** between the potentially misleading caption and the entire factual baseline provided by the article.
>
> **The Article as Necessary Evidential Ground-Truth**
>
> The article context serves as the absolute factual baseline that reveals the semantic discrepancy introduced by the manipulative intent. The performance increase confirms that the VLMs are successfully utilizing the evidential support to perform the **comparison task** (“Is the caption semantically consistent with the factual record of the article?”). This is a core requirement for real-world misinformation detection, which must move beyond self-contained modal analysis.
>
> Alarmingly, even with access to this information-rich evidence, VLMs still perform poorly, particularly on M-Sub cases (Table 4). This validates that the content is still highly challenging, confirming that the models are indeed grappling with the subtle **implication-level mismatch** and not exploiting trivial shortcuts.
>
>
> Therefore, the performance increase is not indicative of "shortcut exploitation", but rather a successful demonstration that models are utilizing the **evidential support** to perform the necessary semantic comparison task inherent to evidence-augmented deception detection.

---

> ### Author Response · Authors · 2025-11-23
> **Rebuttal [5/6]**
>
> **Question 4. In human evaluation, image-modified instances show noticeably lower agreement than text-modified ones (89.2 percent vs 99.2 percent), so could the authors show concrete examples where annotators disagreed to illustrate what types of visual manipulations fail to convey misleading intent clearly.**
>
> **Answer:** We appreciate the reviewer's request to examine the instances of annotator disagreement, as this provides insight into the **subjectivity and subtlety of intent interpretation**, especially for visual modifications. While we achieved substantial agreement for image modifications (Fleiss’ $\kappa = 0.703$), the difference compared to text is informative.
>
> In total, **26 out of 120 image-modified samples** showed disagreement among the annotators. Importantly, most of these cases were concentrated within the **Misleading (Subtle) class**, confirming that the ambiguity is highest near the threshold of perception.
>
> The breakdown reveals that the vast majority of disagreement stems from the nature of our manipulation:
>
> | Source of Disagreement       | Cases (n=26) | Description & Implication |
> |-----------------------------|--------------|---------------------------|
> | Ambiguity/Open Interpretation | 23           | The manipulation is intentionally subtle, creating an interpretive boundary between illustrative style and intentional exaggeration. This reflects real-world ambiguity. |
> | Annotator Oversight         | 1            | Clear labeling error.     |
> | Minor Data Issues           | 2            | Unintended typographical artifact in the generated image. |
>
>
> The dominant cause, **Ambiguity**, is a direct result of our focus on **subtle implication-level manipulation**, which forces annotators to interpret authorial intent rather than rely on the detection of explicit factual error.
>
> To provide the requested concrete illustration, we describe a stylized example drawn from the M-Sub (Image) category (illustrated in Figure 7 of our manuscript)
>
> | Component                     | Description |
> |------------------------------|-------------|
> | **Source News Context**         | An article reporting on a localized flood event in Mill Valley, California. |
> | **Misleading Image Modification** | The original image shows a car driving through a flooded street. The image modification is subtle, involving two distinct additions: (1) the background is lightly manipulated to include distant pillars of smoke, and (2) the foreground is edited to show two people walking on the road carrying their belongings. |
> | **Annotator Disagreement**       | Two annotators (Label: Non-Misleading): Found the image non-misleading, arguing that the inclusion of people carrying belongings is a common and plausible occurrence during a flood, interpreting the additions as merely illustrative realism. |
> |                              | One annotator (Label: Misleading): Found the image misleading, arguing that the combined additions of smoke (implying fire/widespread destruction) and the displaced individuals constitutes an intentional exaggeration of the severity and scope of the localized flood event. |
>
> This type of disagreement is highly valuable: it confirms that our subtle manipulations operate right at the **threshold of perception**, requiring annotators (and, by extension, VLMs) to engage in **deep inferential reasoning about intent**. These edge cases reflect real-world ambiguity and are crucial for future research in modeling graded or implicit intent.
>
> **We have integrated this illustrative example and detailed analysis into Appendix F.1 and Figure 7 of the revised manuscript.**

---

> ### Author Response · Authors · 2025-11-23
> **Rebuttal [6/6]**
>
> **Question 5. Table 5 shows that adding only one line of external framing hint leads to accuracy changes up to +54.5 or −41.8, so could the authors analyze which components of their prompt template are causing models to overly rely on such hints rather than visual–textual evidence.**
>
> **Answer:** We appreciate the reviewer highlighting the extreme sensitivity of VLM performance to a single line of external framing hint, as demonstrated by the significant performance changes (up to $+54.5\%$ or $-41.8\%$) in **Table 5**. This finding highlights a fundamental **instability** in VLM reasoning that our work effectively exposes.
>
> The inconsistency is surprising because our **Consistency-Oriented (C) Prompt** (detailed in Figure 15) explicitly instructs the model to perform evidence-based comparison: "Has the news piece been intentionally distorted, such that there exists inconsistencies between the image and caption, or between the news piece and the reference article?"
>
> While the prompt clearly instructs the model to rely on visual-textual evidence and check inconsistencies, the sudden appearance and placement of the external framing hint appears to be interpreted by the VLM as an overriding, authoritative meta-instruction or "reliable evidence" shortcut, rather than an additional piece of evidence to be weighed.
>
> **Hypothesized Sources of VLM Over-Reliance**
>
> We hypothesize that the VLM is prioritizing the hint due to a strong **Positional and Instruction Authority Bias**:
>
> * **Positional Bias** (Primacy Effect): Placing the hint at the beginning leverages the Primacy Effect, where the VLM disproportionately weights the initial information provided. This causes the hint to act as a filter through which all subsequent evidence (image, caption, article) is interpreted.
>
> * **Instruction Authority**: The hint is treated as a high-authority meta-instruction dictating the expected output, thus overriding the preceding complex task of cross-modal and cross-source consistency checking.
>
>
> **Implication for VLM Reasoning**
>
> This finding is not a flaw in our dataset design but **a key contribution about VLM behavior.** The dramatic sensitivity confirms that the models:
>
> * **Lack Robust Reasoning Structure**: VLMs fail to enforce the logical priority defined by the prompt (i.e., first check inconsistencies, then conclude).
>
> * **Are Highly Susceptible to Shortcuts**: They prioritize the easiest, highest-authority cue (the initial hint) to avoid the complex and computationally expensive task of analyzing multimodal, evidential inconsistencies.
>
> This confirms that the models are not yet capable of stable, evidence-based intent detection, and their performance is **highly brittle**.
>
> **We have updated Section 4.3 of the revised manuscript to include this detailed analysis.**

---

> ### Author Response · Authors · 2025-11-27
> **Gentle reminder of the author-reviewer discussion deadline**
>
> Dear Reviewer fbiN:
>
> We are eager to engage in further discussions with you! In our earlier responses, we have actively addressed your concerns by providing:
>
> 1. Analysis of the **intent fidelity and complexity** within DeceptionDecoded
>
> 2. Explanation of **why potential generator-specific patterns or stylistic artifacts does not influence evaluation**
> 3. **Discussion of the trivial behavior exhibited by some models** that predict nearly all samples as NM (the “NM Bias”)
>
> 4. Empirical results validating the **clear boundary between subtle/significant manipulations** in DeceptionDecoded
>
> 5. **Evidence that style-related bias was removed** in the unreliable style reframing experiment in Figure 3
>
> 6. Clarification on the **central role of news articles in DeceptionDecoded evaluation**, in that they serve as necessary evidence
>
> 7. Insights into **disagreements cases** in human evaluation of misleading intent
>
> 8. Explanation **why the external framing hint affects accuracy** (despite the model being explicitly instructed to utilize visual-textual evidence)
>
> As the discussion deadline is approaching, if you have any additional questions or concerns about the paper, we would be delighted to continue the conversation with you. We sincerely hope that our responses have effectively addressed your concerns and may encourage a more favorable reconsideration of our paper.
>
> Best regards,
>
> Submission 1711 Authors

---

### Official Review · Reviewer_f8Fr · 2025-10-27

**Soundness:** 2
**Presentation:** 2
**Contribution:** 2
**Rating:** 4
**Confidence:** 4

**Summary:**

This paper introduces DECEPTIONDECODED, a benchmark dataset for analyzing misleading creator intent in multimodal news. The dataset contains 12,000 image–caption–article triplets, each grounded in verified VisualNews articles, with both misleading and non-misleading variants generated under predefined “creator intents.”  They evaluate 14 vision–language models, including GPT-4o, Claude-3.7, Gemini-2.5-Pro, and Qwen2.5-VL. The results indicate that even state-of-the-art models perform poorly on intent reasoning, tending to rely on surface-level cues such as image-text consistency or stylistic polish.

**Strengths:**

1. The paper moves beyond conventional multimodal misinformation benchmarks that emphasize factual misalignment, by explicitly modeling creator intention—a rarely addressed but crucial dimension in understanding real-world deception.
2. The cross-model comparison offers valuable insights into the current limitations of multimodal LLMs, especially regarding consistency-based reasoning versus implication-based reasoning.

**Weaknesses:**

1. Diversity of generated intents is insufficiently substantiated. The paper relies heavily on GPT-4o to simulate creator intents but does not provide a systematic method to ensure semantic diversity beyond domain coverage. Merely prompting GPT-4o may not guarantee varied “intent expressions,” leading to possible homogeneity in narrative framing.
2. Exclusion of deepfake/manipulated-identity content limits ecological validity. Identity-based manipulations (e.g., political figures’ face swaps) represent a significant real-world threat vector. The absence of such cases restricts the benchmark’s relevance to the broader multimodal deception landscape.
3. Lack of evaluation for misleading vs. non-misleading cases. Since the distribution is potentially skewed, reporting only accuracy can be misleading. The paper should provide F1 or macro-F1 scores to account for imbalance, and compare with related datasets such as NewsCLIPpings and MMFakeBench to show whether DECEPTIONDECODED introduces greater realism or challenge.
4. Ambiguity in controlling the boundary between subtle and significant manipulations. The paper distinguishes between “subtle” and “significant” misleading cases in both image and text manipulations, but it is unclear how the generation process enforces or verifies this distinction quantitatively. Moreover, the reported 2% human evaluation is used only to assess label consistency and plausibility, which is insufficient to guarantee that the subtle–significant boundary is consistently maintained across 12,000 samples.

**Questions:**

5. Ambiguity between the “implication-oriented (I)” and “consistency-oriented (C)” paradigms.
While both evaluation paradigms are central to the analysis, their conceptual distinction and implications for model reasoning are not clearly defined.

---

> ### Author Response · Authors · 2025-11-23
> **Rebuttal [1/5]**
>
> Thanks for your feedback. We’re pleased that you found our investigation of creator intent of high real-world relevance, and our experiments of valuable findings..
>
> We hope our point-by-point responses addresses your concerns:
>
> ----------------------
>
> **Weakness 1. Diversity of generated intents is insufficiently substantiated. The paper relies heavily on GPT-4o to simulate creator intents but does not provide a systematic method to ensure semantic diversity beyond domain coverage. Merely prompting GPT-4o may not guarantee varied “intent expressions,” leading to possible homogeneity in narrative framing.**
>
> **Answer:** Thanks for raising the point about ensuring deep semantic diversity, which is fundamental to moving beyond simple domain coverage and preventing narrative homogeneity. We agree that relying solely on a simple, naive prompt to GPT-4o would risk producing trivial or limited intent expressions.
>
> Our approach systematically substantiates semantic diversity through three layered mechanisms: a principled generation framework, empirical distribution analysis, and human validation.
>
> **Principled, Fine-Grained Intent Formalization**
>
> The core of our method lies in the Intent-Guided Misinformation Simulation Framework (Section 3.2), which operationalizes strategic communication theory. This structure forces GPT-4o to explore a combinatorial semantic space, systematically guaranteeing variation. Specifically, Each misleading instance is decomposed into two theoretically grounded, fine-grained components:
>
> * **Desired Influence**: This is defined by a list of **8 fine-grained, action-oriented influence targets** (e.g., Public Health and Safety issues, economic misinformation). GPT-4o is explicitly conditioned to **choose and integrate 2-3** of these targets per instance, forcing complex, layered narrative framing.
>
> * **Execution Plan**: Execution Plan: This specifies the **rhetorical and structural strategy** (e.g., selective omission, misrepresentation of statistics), ensuring the intent is expressed through diverse textual and visual manipulations.
>
> This structure moves far beyond “mere prompting”, leveraging the LLM to combine established strategies, thereby ensuring deep semantic variation in how the final intent is expressed.
>
> **Empirical Verification of Intent Distribution**
>
> The resulting benchmark, DeceptionDecoded, empirically exhibits a highly diverse distribution across our 8 Intent Types, confirming the framework's effectiveness:
>
>
> | Intent Type                            | M-Sub (Text) | M-Sig (Text) | M-Sub (Image) | M-Sig (Image) |
> | -------------------------------------- | ----- | ----- | ----- | ----- |
> | Political Polarization                 | 581   | 732   | 462   | 566   |
> | Social Polarization                    | 467   | 517   | 490   | 519   |
> | Cultural and Religious Polarization    | 29    | 27    | 19    | 9     |
> | Economic Misinformation                | 309   | 268   | 272   | 191   |
> | Public Health and Safety               | 913   | 942   | 1,011 | 1,096 |
> | Environmental and Scientific Polarization    | 467   | 430   | 531   | 509   |
> | Geopolitical and International Relations | 319   | 501   | 305   | 390   |
> | Psychological and Emotional Manipulation  | 1,744 | 1,911 | 1,886 | 1,944 |
>
>
> Observation: This distribution **mirrors patterns documented in prior misinformation research** [1, 2], with “Psychological and Emotional Manipulation” and “Public Health and Safety” being most common, demonstrating that the LLM is following real-world salience rather than defaulting to a uniform or biased output. Crucially, the presence of each intent type across all four manipulation categories (subtle/signature, text/visual) confirms that intents are expressed in **varied modalities and transformation styles**.
>
> **We have discussed this  in detail in Appendix F.1 of our revised manuscript, highlighted in blue.**
>
> **Human Validation of Fidelity**
>
> Finally, as detailed in **Appendix F.2 and F.3**, we conducted a human evaluation specifically assessing **Intent Fidelity** and **News Plausibility**. High agreement scores across all domains and intent categories empirically confirm that the generated intents are not only diverse in structure but are also **coherent and realistic expressions of persuasive strategies**.
>
> The combination of this theoretically rigorous, combinatorial generation method with empirical verification and human validation ensures the deep semantic diversity required for a high-quality benchmark.
>
> [1] Mind Over Media: Propaganda Education for a Digital Age. Renee Hobbs, 2020.
>
> [2] The disaster of misinformation: a review of research in social media. In International Journal of Data Science and Analytics, 2022.

---

> ### Author Response · Authors · 2025-11-23
> **Rebuttal [2/5]**
>
> **Weakness 2. Exclusion of deepfake/manipulated-identity content limits ecological validity. Identity-based manipulations (e.g., political figures’ face swaps) represent a significant real-world threat vector. The absence of such cases restricts the benchmark’s relevance to the broader multimodal deception landscape.**
>
> **Answer:** We appreciate your feedback on our scope definition and the important point regarding ecological validity. We agree that identity-based manipulations (e.g., political figure face swaps) represent a significant threat vector where manipulation can be detected by analyzing **explicit factual consistency**.
>
> Our initial design, however, strategically prioritized a different, equally prevalent, and arguably more insidious form of deception: **implication-level shifts** that maintain surface-level factual consistency. Our focus is on cases where: **The underlying event and participants remain unchanged, but subtle edits to the image or text shift the interpretation of the news item.** Psychological studies [3, 4] confirm that these small, implication-level edits significantly alter reader memory and inference.
>
> We consider this implication-level mismatch to be **strategically trickier** for models (and humans) than explicit factual mismatch (like an obvious face swap), as the manipulation cannot be detected by simply checking factual alignment. The difficulty lies in reasoning about subtle intent, which is the primary focus of our benchmark.
>
> **Generalization through Intent-Guided Image Editing**
>
> Nevertheless, we utilized the image editing perspective raised in your comment to demonstrate the **broader utility and ecological relevance** of our intent-guided data generation framework. We conducted a new pilot study to show that the framework generalizes effectively to realistic high-fidelity, localized image editing: **a technique closely related to identity manipulation.**
>
> * **Framework Application**: We adapted our framework (Section 3.1 & 3.2) to guide **high-fidelity localized editing** instead of full image generation. This involved establishing the creator’s communicative intent (e.g., increase perceived threat) and using a VLM to execute precise, intent-aligned edits (e.g., altering background lighting to look menacing, or modifying subtle scene elements).
>
> * **Generation**: We employed **Google's Nano Banana model** (gemini-2.5-flash-image) to generate **400 new edited samples** (Subtle and Significant Image Distortions), following the pilot setup in Section 5.2 of our submission.
>
> * Benchmarking Results: We tested five VLMs using our standard detection setup (Figure 15):
>
>
> | Model           | M-Sub (Image) GPT-gen | M-Sub (Image) Edited | M-Sig (Image) GPT-gen | M-Sig (Image) Edited |
> |----------------|------------------------|-----------------------|------------------------|-----------------------|
> | GPT-4o         | 66.0                   | 48.0                  | 84.0                   | 61.0                  |
> | GPT-4o-mini    | 47.5                   | 22.5                  | 61.5                   | 41.5                  |
> | Claude-3.5-Haiku | 9.0                  | 6.0                   | 18.0                   | 11.5                  |
> | Qwen2.5-72B    | 53.5                   | 32.0                  | 57.0                   | 52.5                  |
> | Qwen2.5-32B    | 37.0                   | 21.5                  | 46.0                   | 41.0                  |
>
>
>
> The results are highly informative and reinforce the relevance of our approach:  (1) Models are **consistently and significantly weaker** at detecting visually conveyed deception in the precisely edited samples than in the fully generated images. (2) The high difficulty of generated image edits demonstrates that **our intent-guided simulation framework naturally extends** to diverse and challenging manipulation types like fine-grained editing. (3) This finding highlights the severe and immediate risks associated with intent-driven, localized visual manipulation capabilities in models like Nano Banana, confirming the practical relevance of our study.
>
> This confirms that our intent-guided framework successfully creates ecologically valid and challenging manipulations that generalize across different visual synthesis techniques. **We have included this pilot study in Section 5.3 of the revised manuscript, complete with illustrative examples, to broaden the scope and address the ecological validity concern.**
>
> [3] The effects of subtle misinformation in news headlines. In Journal of Experimental Psychology: Applied, 2014.
>
> [4] Epistemic language in news headlines shapes readers’ perceptions of objectivity. In PNAS 2024.

---

> ### Author Response · Authors · 2025-11-23
> **Rebuttal [3/5]**
>
> **Weakness 3. Lack of evaluation for misleading vs. non-misleading cases. Since the distribution is potentially skewed, reporting only accuracy can be misleading. The paper should provide F1 or macro-F1 scores to account for imbalance, and compare with related datasets such as NewsCLIPpings and MMFakeBench to show whether DECEPTIONDECODED introduces greater realism or challenge.**
>
> **Answer:** Thanks for your suggestion regarding evaluation metrics and the request for a comparative analysis, which helps contextualize the challenge posed by DeceptionDecoded.
>
> **Evaluation Metrics and Dataset Balance**
>
> We would like to reassure the reviewer that DeceptionDecoded is perfectly balanced across its six fundamental classes (**Lines 238-242**). Specifically, The dataset comprises **12,000 samples** distributed equally across the six fine-grained classes (Misleading-Subtle Text, Misleading-Significant Text, Non-Misleading Text, and their three Image modification counterparts), with **2,000 samples per class**. For a granular view of performance, we already report the **class-wise performance breakdown** in Table 2. To assess overall detection capability per modality, we compute the **macro-accuracy** over the three classes within that modality (Subtle, Significant, Non-Misleading), effectively accounting for the diverse degrees of manipulation.
>
> **Comparison with Related Datasets (Challenge and Realism)**
>
>
> To validate that DeceptionDecoded introduces greater realism and challenge, we compare it against NewsCLIPpings, which is the most directly comparable dataset, as it focuses on detecting out-of-context misinformation using image-caption pairs derived from real news. We contrast the **factual mismatch task (NewsCLIPpings)** with **our implication-level mismatch task.**
>
> We ensure a fair comparison by benchmarking GPT-4o's performance in both datasets using an **evidence-augmented reasoning approach** (evidence refers to available article/context). Specifically, we compare (1) the consistency-oriented detection performance reported in Table 2 of our work, with (2) an ablated approach reported in CMIE [5] on NewsCLIPpings, where the model is provided with external evidence.
>
> | Dataset             | Detection Task                                   | GPT-4o Accuracy (%) |
> |---------------------|---------------------------------------------------|-----------------------|
> | NewsCLIPpings   | Detecting out-of-context images (factual mismatch)   | 85.0%   [3]              |
> | DeceptionDecoded    | Detecting intent-driven manipulation (implication-level mismatch) | 74.85%              |
>
>
> **The results clearly show that our intent-driven, implication-level manipulation makes DeceptionDecoded significantly more challenging than detecting out-of-context misinformation, with a performance drop of over 10%.** This validates that our novel framework successfully introduces greater realism and highlights a pressing challenge in the multimodal deception landscape.
>
>
>
> [5] CMIE: Combining MLLM Insights with External Evidence for Explainable Out-of-Context Misinformation Detection. In ACL 2025 Findings.

---

> ### Author Response · Authors · 2025-11-23
> **Rebuttal [4/5]**
>
> **Weakness 4. Ambiguity in controlling the boundary between subtle and significant manipulations. The paper distinguishes between “subtle” and “significant” misleading cases in both image and text manipulations, but it is unclear how the generation process enforces or verifies this distinction quantitatively. Moreover, the reported 2% human evaluation is used only to assess label consistency and plausibility, which is insufficient to guarantee that the subtle–significant boundary is consistently maintained across 12,000 samples.**
>
> **Answer:** Thanks for pointing out the ambiguity in controlling and verifying the critical distinction between subtle and significant manipulations. Maintaining this controlled boundary is important to ensure that DeceptionDecoded serves as a true test of a model's sensitivity to varying degrees of deception.
>
> We enforce and validate this distinction using a multi-pronged approach that combines explicit procedural guidance with large-scale empirical validation:
>
> **Explicit Procedural Guidance in Generation**
>
> The distinction between Subtle and Significant manipulation is enforced through explicit, differentiated instructions that guide the LLM's strategy. These instructions demand perceptually distinct outcomes:
>
>
> | Manipulation Degree | Prompt Snippet Example | Strategic Goal |
> |---------------------|------------------------|----------------|
> | Subtle | “Retains key elements of the original while **subtly reframing** the context. Slightly manipulates audience perception while remaining credible and convincing.” | Targets implication-level shifts, maintaining high plausibility. |
> | Significant | “**Drastically distorts** the original, while maintaining basic alignment with key elements portrayed in the news context.” | Targets obvious shifts in meaning, while still avoiding explicit factual contradiction with the original source. |
>
> These instructions translate the conceptual difference into specific, measurable generation goals for the LLM.
>
>
> **Large-Scale Empirical Validation**
>
> To verify the perceptual difference at scale, we conducted an **empirical validation** using the (Multimodal) LLM-as-a-Judge methodology, which serves as a powerful, cost-effective proxy for human perception across the entire dataset's complexity:
>
> * Experiment: We randomly sampled **500 original news items**. We then fed the corresponding Subtle and Significant variants (in randomized order) into a powerful external model (GPT-4.1) and asked it to judge which version exhibited the **higher degree of distortion.**
>
> * Results: The **high win rates of Significant over Subtle** validate the success of our design: **Text: 99.4%, Image: 97.6%**
>
> These results confirm that our procedural guidance yields a high inter-class fidelity, ensuring that the Subtle and Significant classes represent two distinct perceptual difficulties necessary for evaluating model sensitivity.

---

> ### Author Response · Authors · 2025-11-23
> **Rebuttal [5/5]**
>
> **Question 1: Ambiguity between the “implication-oriented (I)” and “consistency-oriented (C)” paradigms. While both evaluation paradigms are central to the analysis, their conceptual distinction and implications for model reasoning are not clearly defined.**
>
> **Answer:** We appreciate the request for a clearer conceptual distinction between the Implication-Oriented (I) and Consistency-Oriented (C) paradigms, and their respective implications for model reasoning. These two evaluation approaches are central to isolating different dimensions of VLM intelligence when faced with deception.
>
> The core difference lies in the **type of violation** the model is instructed to detect:
>
> **Implication-Oriented Paradigm (I)**
>
> The Implication-Oriented paradigm focuses the model purely on the inferred manipulative effect (the intent).
>
> * **Core Question**: "Has the news piece been intentionally distorted by its creator to mislead readers, such that it conveys a biased, controversial, or manipulated message?" (Illustrated in Figure 14).
>
> * **Conceptual Distinction**: The model is asked to judge the **Intentional Integrity** of the message. It must infer the **creator's intent** based on the overall tone and framing, even if a concrete factual contradiction is not present.
>
> * **Implication for Model Reasoning**: This paradigm enforces **Holistic, Inference-Based Reasoning**. The model must rely on subtle cues and rhetorical framing to determine if the implication of the message is distorted. This is beneficial for detecting our **Subtle manipulations** where factual consistency is high.
>
>
> **Consistency-Oriented Paradigm (C)**
>
> The Consistency-Oriented paradigm directs the model to perform explicit checks against available evidence.
>
> * **Core Question**: "Has the news piece been intentionally distorted by its creator to mislead readers, such that it conveys a biased, controversial, or manipulated message?" (Illustrated in Figure 14).
>
> * **Conceptual Distinction**: The model is asked to judge **Factual/Implication-Level Integrity** by verifying cross-modal and cross-source alignment. The focus shifts from inferred intent to verifiable contradiction.
>
> * **Implication for Model Reasoning**: This paradigm enforces **Evidence-Based Comparison**. The model must perform explicit cross-modal checks (image vs. caption) and cross-source checks (pair vs. article). This is particularly effective for detecting Significant manipulations, where the distortion is larger and more likely to result in a measurable inconsistency.
>
> **Interpretation of Performance**
>
>
> As shown in **Table 2**, consistency-oriented reasoning (C) generally outperforms implication-oriented reasoning (I). Our interpretation is that models find **detecting an unsubstantiated inconsistency (a failure in eithwr factual or implication-level integrity)** to be a more reliable signal for identifying misleading intent than solely relying on difficult, holistic inference (intentional integrity). This highlights that current VLMs are better at performing comparison against evidence than they are at inferring intent from subtle rhetorical cues.

---

> ### Author Response · Authors · 2025-11-23
> **Response to Flag For Ethics Review**
>
> **Flag For Ethics Review: Yes, Privacy, security and safety.**
>
> **Answer:** We deeply appreciate the attention to the ethical dimensions of this work, especially regarding the risks of Privacy, security, and safety. We agree that transparently addressing potential misuse is an essential responsibility when reporting empirical insights on the vulnerability of advanced models to realistic, intent-guided deception.
>
> As detailed in our Ethics Statement (**Appendix B** of our submission), our project is designed to maximize the scientific benefit (understanding model vulnerabilities) while minimizing the risk of real-world harm. Our primary contribution is exposing the vulnerability of existing VLMs to sophisticated manipulation, which is essential for developing robust defenses. We have implemented several layered safeguards:
>
> **Mitigation of Misuse (Security and Safety)**
>
> The greatest risk lies in weaponizing our intent-guided framework for the large-scale generation of convincing misleading content. To counter this, we have adopted stringent security measures, as outlined in **Appendix B** in original submission:
>
> * **Non-Release of Generator Prompts**: We will not open-source the specific, high-fidelity generation prompts (the instruction set used to guide GPT-4o and FLUX) that could be repurposed for deception.
>
> * **Restricted Access**: Access to the DeceptionDecoded dataset and evaluation scripts will be strictly limited to verified academic and institutional researchers under a binding Data Usage Agreement. This explicitly prohibits the use of the data or framework for any harmful or offensive purposes.
>
> **Privacy and Identifiability Safeguards**
>
> To prevent potential harm or privacy infringement involving real people, we have established clear guidelines for the content itself:
>
> * **Anonymization of Entities**: All simulated news content is based on anonymized, public events or utilizes non-identifiable entities.
>
> * **Focus on Concepts, Not Individuals**: We have explicitly excluded the manipulation of news content that involves specific, identifiable political figures or private individuals.
>
> These steps: restricting access, withholding the generative methodology, and anonymizing the content, form the ethical framework that we committed to in Appendix B of the original submission.
>
> **We are committed to continuing this discussion and are ready to address any further/specific concerns.**

---

> ### Author Response · Authors · 2025-11-27
> **Gentle reminder of the author-reviewer discussion deadline**
>
> Dear Reviewer f8Fr:
>
> We are eager to engage in further discussions with you! In our earlier responses, we have actively addressed your concerns by providing:
>
> 1. Analysis on the **diversity of generated intents**
>
> 2. An extension of our intent-guided misinformation simulation framework to **image editing**
>
> 3. Clarification of **class balance in our data and evaluation metric selection**
>
> 4. Empirical results validating the **clear boundary between subtle/significant manipulations** in DeceptionDecoded
>
> 5. **Conceptual distinction and implications of implication- and consistency-oriented paradigms** for model reasoning
>
> 6. Additional clarification of **ethical considerations**, building on our Ethics Statement in Appendix B
>
> As the discussion deadline is approaching, if you have any additional questions or concerns about the paper, we would be delighted to continue the conversation with you. We sincerely hope that our responses have effectively addressed your concerns and may encourage a more favorable reconsideration of our paper.
>
> Best regards,
>
> Submission 1711 Authors

---

### Official Review · Reviewer_cUUE · 2025-10-31

**Soundness:** 2
**Presentation:** 2
**Contribution:** 2
**Rating:** 4
**Confidence:** 3

**Summary:**

This paper introduces DECEPTIONDECODED, a novel benchmark designed to evaluate Vision-Language Models (VLMs) in detecting creator intent behind misleading multimodal news content. The dataset is constructed using a synthetic, intent-guided framework that generates manipulations grounded in real news, ensuring relevance and control over deception intent. The study evaluates state-of-the-art VLMs under various input conditions (e.g., image+text, text+article) and with authenticity cues (helpful or adversarial hints).

**Strengths:**

1 The paper shifts the focus from simple fact-checking to intent detection in multimodal misinformation, which is a more nuanced and realistic challenge. This addresses a critical gap in current VLM evaluation.

2 DECEPTIONDECODED is well-constructed, grounded in real news, and systematically manipulates intent. The use of human evaluation to validate realism, intent alignment, and label correctness strengthens the dataset’s credibility.

3 The experimental design is thorough, testing models under multiple input modalities and with adversarial authenticity cues. This provides deep insights into model behavior and failure modes.

4 The findings reveal specific weaknesses in current VLMs—such as susceptibility to misleading images, reliance on style over substance, and vulnerability to spurious hints—offering clear directions for future research.

**Weaknesses:**

1 The paper is purely diagnostic. It identifies problems but does not propose or evaluate any methods to improve VLMs’ robustness against the identified deception strategies, limiting its impact on advancing model capabilities.

2 Although inter-annotator agreement is measured, judgments about “misleading intent” can be subjective. The paper could better discuss edge cases or ambiguous examples where annotators disagreed.

3 The study focuses primarily on caption manipulation within a fixed image-news context. Real-world misinformation often involves deeper fabrications, manipulated images, or entirely synthetic content, which are not addressed.

**Questions:**

1 The paper is purely diagnostic. How can the insights from this benchmark be used to build more resilient models? What concrete methods do the authors suggest for improving model robustness using these findings?

2 While inter-annotator agreement is reported, what types of manipulations led to disagreement on “misleading intent”? Could you discuss a few ambiguous cases to better illustrate the subjectivity in labeling?

3 The study focuses on caption manipulation with real images. How might the findings change if the images themselves were manipulated or synthetic, as is common in real-world disinformation?

---

> ### Author Response · Authors · 2025-11-23
> **Rebuttal [1/3]**
>
> Thanks for your feedback. We’re pleased that you found our intent-oriented perspective important and realistic, our DeceptionDecoded benchmark well-designed and validated, our experiments thorough, and our findings valuable for inspiring future research.
>
> We hope our point-by-point responses addresses your concerns:
>
> ----------------------
>
> **Weakness/Question 1: The paper is purely diagnostic. It identifies problems but does not propose or evaluate any methods to improve VLMs’ robustness against the identified deception strategies, limiting its impact on advancing model capabilities.**
>
> **Answer:** Thanks for raising this point. While our study is fundamentally **diagnostic** in its evaluation of current VLM capabilities, we argue that its contribution is also significantly **constructive**, directly enabling the development of more robust deception detectors.
>
> Our work moves beyond problem identification by providing both the **data** and the **methodology** necessary to advance model capabilities against intent-driven deception.
>
> **Constructive Contribution: The Intent-Guided Synthesis Framework**
>
> The central technical contribution of our work is the **automated intent-guided news creation framework (Section 3)**. This framework operationalizes strategic communication theories to synthesize **high-fidelity, article-grounded multimodal news**. This framework directly addresses the core barrier to developing resilient misinformation detectors: the lack of high-quality, intent-focused multimodal training data.
>
> * **Scalable Data Solution**: Because understanding misleading intent is essential for reliable content moderation, our synthesis pipeline **provides a principled, scalable way** to model and reproduce realistic deception strategies. This methodology enables the creation of vast amounts of targeted training data, not just static diagnostic samples.
>
>
> **Empirical Evidence: Enhancing Model Robustness**
>
> To demonstrate this utility for model improvement, we presented a transfer experiment in **Section 5.1 of our submission**. We fine-tuned two representative base VLMs on a subset of DeceptionDecoded samples and evaluated their performance on MMFakeBench, a well-known benchmark for general multimodal misinformation detection.
>
> As shown in **Table 6 (copied below)**, fine-tuning yields consistent and **significant gains** across both Macro-F1 and Macro-Accuracy:
>
>
> | Model                     | Macro-F1 (%)          | Macro-Acc. (%)         |
> |---------------------------|------------------------|-------------------------|
> | Llava-v1.6-7B             | 44.41                  | 50.31                   |
> | + FFT (w/ 6k samples)     | **52.37 (+7.96)**      | **63.52 (+13.21)**      |
> | Qwen2.5-VL-7B             | 27.96                  | 52.12                   |
> | + FFT (w/ 6k samples)     | **58.66 (+30.70)**     | **67.24 (+15.12)**      |
>
>
> The substantial performance gains (up to +30.70% Macro-F1) on an independent, diverse dataset demonstrate that DECEPTIONDECODED successfully instills a generalized ability to detect intent-based deception.
>
> In this sense, our contribution is dual: the benchmark provides systematic **evaluation** (diagnosis), and the underlying framework provides the **data generation strategy** (construction) necessary for practical enhancement of VLMs against realistic, intent-driven multimodal deception.

---

> ### Author Response · Authors · 2025-11-23
> **Rebuttal [2/3]**
>
> **Weakness/Question 2: Although inter-annotator agreement is measured, judgments about “misleading intent” can be subjective. The paper could better discuss edge cases or ambiguous examples where annotators disagreed.**
>
> **Answer:** Thank you for this helpful observation. While Fleiss’ $\kappa$ = 0.703 for image and  Fleiss’ $\kappa$ = 0.877 for text indicates substantial agreement among the three annotators (Section 3.4), we agree that examining disagreement cases provides deeper insight into the subjectivity and subtlety of intent interpretation.
>
> To better understand the ambiguities, we analyzed DeceptionDecoded cases where annotators disagreed on whether a modified multimodal news item conveyed misleading intent.
>
> **Analysis of Disagreement Sources**
>
> Across the dataset, annotators disagreed on **26 out of 120 image-modified samples** and **10 out of 120 caption-modified samples**. Notably, the majority of these disagreements occurred within the Misleading (Subtle) class, suggesting that ambiguity is concentrated in subtle manipulations rather than those with clearer deceptive purposes.
>
> A qualitative review reveals three primary sources of disagreement:
>
> * Intent Ambiguity (Dominant Cause): The modification is deliberately mild, blurring the line between stylistic emphasis and intentional exaggeration. This mirrors real-world ambiguity in creator intent.
>
> * Annotator Oversight: cases of clear labeling mistakes rather than genuine interpretive differences.
>
> * Data Generation Issues: Rare artifacts introduced during content generation (e.g., visual or typographical anomalies) led to confusion unrelated to the intended deception.
>
> The distribution of edge cases is shown as follows:
>
> | Category                  | Intent Ambiguity | Annotator Oversight | Data Generation Issues |
> |--------------------------|------------------|----------------------|-------------------------|
> | Image Modifications (n=26) | 23               | 1                    | 2                       |
> | Text Modifications (n=10)  | 10               | 0                    | 0                       |
>
> **Illustrative Edge Case: Intent Ambiguity**
>
> The dominant source of disagreement arises from **Intent Ambiguity**. These edge cases reflect real-world ambiguity in how creators embed misleading implications at the threshold of perception.
>
> For instance, in the **M-Sub (Image)** category, annotators disagreed on a sample where the misleading intent was to exaggerate the scope of a localized flood event (illustrated in **Figure 7** of revised manuscipt):
>
> * **Source Context**: Trustworthy article about a localized flood in a small town.
>
> * **Misleading Image Modification**: The image shows a car driving through a flooded street. The manipulation involved **subtly adding distant pillars of smoke and two people walking with their belongings** in the foreground.
>
> * **Disagreement**: Two annotators scored the image Non-Misleading, viewing the additions as plausible illustrative details common in disaster reporting. The third annotator scored it Misleading, interpreting the smoke and displaced people as an intentional, subtle exaggeration of the disaster's severity beyond what was factually reported.*
>
> This type of disagreement is highly valuable as it confirms that our subtle manipulations operate right at the threshold of perception, requiring annotators to engage in deep inferential reasoning about authorial intent.
>
> **We have incorporated this discussion, including detailed figures (e.g., Figure 7-10 for specific examples) and the summary table (Table 10), into Appendix F.1 of the updated manuscript.** This strengthens the transparency and interpretability of our annotation process and highlights the specific challenges for future work on graded intent modeling.

---

> ### Author Response · Authors · 2025-11-23
> **Rebuttal [3/3]**
>
> **Weakness/Question 3: The study focuses primarily on caption manipulation within a fixed image-news context. Real-world misinformation often involves deeper fabrications, manipulated images, or entirely synthetic content, which are not addressed.**
>
> **Answer:** We appreciate the opportunity to clarify the scope of manipulations investigated in our benchmark.
>
> **There appears to be a misunderstanding: DeceptionDecoded is not limited to caption manipulation.** Our original work already covers **both image and text modification**, focusing on intent-guided, implication-level shifts in a single modality.
>
> **Clarification of Original Scope**
>
> Our initial dataset design (detailed in **Section 3.3**) already includes two main classes of deception:
>
> * **Caption Modification (Text-Modified)**: The original, authentic image is kept, and the caption is reframed to convey a misleading implication.
>
> * **Image Modification (Image-Modified)**: The original, authentic caption is kept, and the image is entirely synthesized to visually align with the caption while misleading readers (e.g., by exaggerating a scenario).
>
> **Extension to Fine-Grained Image Editing**
>
> To directly address your suggestion regarding **deeper, localized fabrications and fine-grained editing**, we utilized our intent-guided framework to conduct a new pilot experiment incorporating advanced image editing technology.
>
> * **Intent-Guided Editing**: We extended our pipeline to generate detailed, intent-aligned instructions for precise edits (e.g., adding localized smoke, changing object context), and used **Google's Nano Banana model** (gemini-2.5-flash-image) to execute the modifications.
>
> * **New Samples**: We created **400 new edited samples** across the M-Sub (Image) and M-Sig (Image) classes.
>
> The results confirm that these fine-grained edits pose an even greater challenge to detection:
>
> | Model           | M-Sub (Image) GPT-gen | M-Sub (Image) Edited | M-Sig (Image) GPT-gen | M-Sig (Image) Edited |
> |----------------|------------------------|-----------------------|------------------------|-----------------------|
> | GPT-4o         | 66.0                   | 48.0                  | 84.0                   | 61.0                  |
> | GPT-4o-mini    | 47.5                   | 22.5                  | 61.5                   | 41.5                  |
> | Claude-3.5-Haiku | 9.0                  | 6.0                   | 18.0                   | 11.5                  |
> | Qwen2.5-72B    | 53.5                   | 32.0                  | 57.0                   | 52.5                  |
> | Qwen2.5-32B    | 37.0                   | 21.5                  | 46.0                   | 41.0                  |
>
>
> These results lead to three key conclusions: (1) Models are **consistently and significantly weaker** at detecting visually conveyed deception in the precisely edited samples than in the fully generated images. (2) The high difficulty of generated image edits demonstrates that **our intent-guided simulation framework naturally extends** to diverse and challenging manipulation types like fine-grained editing. (3) This finding highlights the severe and immediate risks associated with intent-driven, localized visual manipulation capabilities in models like Nano Banana, confirming the practical relevance of our study.
>
> **We have included the full details of this extended study in Section 5.3 of the revised manuscript.**

---

> > ### Comment · Reviewer_cUUE · 2025-11-26
> > **I will keep my score**
> >
> > The reviewer clarified my questions quite clearly, but the biggest problem with the paper is that it does not propose a method to enhance the robustness of VLMs against the identified deceptive strategies, which limits its impact on advancing model capabilities. This issue has not been well addressed in the current version.

---

> > > ### Author Response · Authors · 2025-11-27
> > > **Response to Follow-up Concern: Missing Method for Enhancing Model Robustness [1/2]**
> > >
> > > Thank you for sharing your remaining concern. We are glad that our earlier clarifications were helpful.
> > >
> > > --------------------
> > >
> > > **Follow-Up Question:** The biggest problem with the paper is that it does not propose a method to enhance the robustness of VLMs against the identified deceptive strategies, which limits its impact on advancing model capabilities.
> > >
> > > **Answer:** We appreciate your continued focus on the important challenge of **advancing VLM capabilities**. We agree that a diagnostic benchmark must also lead to concrete solutions. Our work addresses this through **(1) a methodological solution**: our intent-guided simulation framework, and **(2) new empirical evidence** that this solution effectively enhances VLM robustness against intent-driven deception.
> > >
> > > We understand that the perceived “missing method” gap arises from viewing the framework as diagnostic only. In fact, **the framework is the method that enables scalable, high-fidelity data generation tailored to misleading intent**, which is precisely the missing resource that has prevented progress on this problem.
> > >
> > > **1. The Intent-Guided Framework: A Method for Enhancement**
> > >
> > > Our core technical contribution is the **intent-guided multimodal news simulation framework** (Section 3.2) This framework is not just for diagnostics; it serves as the constructive method for enhancement:
> > >
> > > * Problem: The lack of efforts and data focused on **misleading intent** limits VLM robustness to deception.
> > >
> > > * Solution (The Method): Our framework systematically synthesizes data that compels models to learn **implication-level intent reasoning**: the capability we clearly identified as central to effective multimodal misinformation detection (MMD) in Abstract and Introduction.
> > >
> > > **2. Empirical Validation: Fine-Tuning and Broader Transferability**
> > >
> > > To demonstrate that our method enhances VLM capabilities, we provide two empirical evaluations showing the framework’s direct impact on robustness to deception.
> > >
> > > **A. Acquisition of Core Intent Detection Capability**
> > >
> > > We first show that **full fine-tuning on DeceptionDecoded enables VLMs to acquire the specific ability to detect misleading intent**. Using the same misleading-intent detection template as in our consistency-oriented prompt (core excerpt in Q1 of Figure 15), we fully fine-tune Qwen2.5VL-7B on 80% of our dataset (9.6K samples derived from 1.6K source news items, evenly distributed across 10 domains) and evaluate on the held-out 20% (2.4K samples derived from 400 unseen source items, also evenly distributed across the same domains).
> > >
> > > Compared with the original Qwen-2.5VL-7B performance:
> > >
> > > | Model                               | M-Sub (Text) | M-Sig (Text) | NM (Text) | Avg. (Text) | M-Sub (Image) | M-Sig (Image) | NM (Image) | Avg. (Image) |
> > > |-------------------------------------|--------------|---------------|-----------|-------------|----------------|----------------|------------|---------------|
> > > | GPT-4o (Table 2)                    | 70.2         | 93.3          | 86.6      | 83.4        | 51.7           | 69.3           | 78.0       | 66.3          |
> > > | Qwen2.5-VL-7B (Table 2)             | 0.0          | 0.0           | 100.0  | 33.3        | 0.0            | 0.0            | 100.0  | 33.3          |
> > > | Qwen2.5-VL-7B (w/ Full Fine-Tuning) | 78.7     | 95.0      | 94.3      | **89.3**    | 54.8       | 66.0       | 82.8       | **67.9**      |
> > >
> > >
> > > These consistent and substantial gains across all six classes confirm that our data successfully equips the VLM with the **targeted** capability of reasoning about misleading intent—bringing a 7B open model to performance levels comparable to a strong closed-source model such as GPT-4o.

---

> ### Author Response · Authors · 2025-11-27
> **Response to Follow-up Concern: Missing Method for Enhancing Model Robustness [2/2]**
>
> **B. Extended Transferability to Diverse Real-World MMD Benchmarks**
>
> More importantly, we show that **training on our intent-focused data generalizes to broader, real-world MMD**. Building on the transfer experiments in Section 5.1, we evaluate on two additional widely used, real-world MMD benchmarks: **Fakeddit** [1] and **FakeNewsNet** [2]. We sample 1,000 multimodal news items from each benchmark (500 Real and 500 Fake) to construct the corresponding test sets.
>
> Following the same setup as in Section 5.1, we compare VLM performance before and after fine-tuning on 6K samples in DeceptionDecoded:
>
> | Model (Macro-F1 %)        | MMFakeBench        | Fakeddit           | FakeNewsNet        |
> |---------------------------|--------------------|--------------------|--------------------|
> | Llava-v1.6-7B             | 44.41              | 34.63              | 43.73              |
> | + FFT (6k samples)        | **52.37** *(+7.96)* | **39.43** *(+4.80)* | **65.22** *(+21.49)* |
> | Qwen2.5-VL-7B             | 27.96              | 32.69              | 44.87              |
> | + FFT (w/ 6k samples)     | **58.66** *(+30.70)* | **41.96** *(+9.27)* | **68.59** *(+23.72)* |
>
>
>
> These consistent and substantial improvements across three diverse and independent MMD benchmarks validate that:
>
> * Models are learning **generalized principles of deception detection**, not just dataset artifacts.
>
> * Training on **creator-intent focused data**, derived from our intent-guided simulation framework, is an **effective and scalable strategy** for improving robustness to real-world multimodal misinformation.
>
> In conclusion, our work is not just diagnostic. The **intent-guided framework is the proposed method** for enhancing VLM robustness, and the **fine-tuning results are the empirical validation of its impact** on advancing VLM robustness to intent-based deception.
>
> -----
>
> **We have updated Section 5.1 and Appendix G.5.2 of our manuscript to highlight this insight. We sincerely hope that the additional clarifications and results have effectively addressed your remaining concern and may encourage a more favorable reconsideration of our paper.**
>
> [1] r/Fakeddit: A New Multimodal Benchmark Dataset for Fine-grained Fake News Detection. In LREC 2020.
>
> [2] Fakenewsnet: A data repository with news content, social context, and spatiotemporal information for studying fake news on social media. In Big Data, 2020.

---

> > ### Comment · Reviewer_cUUE · 2025-11-28
> > **The model's transferability is surprisingly impressive, and I'm inclined to raise the score.**
> >
> > Although the paper does not propose a new model, its fine-tuning approach demonstrates significant performance improvements across multiple datasets, highlighting the method's strong transferability.

---

> > > ### Author Response · Authors · 2025-11-28
> > > **Thanks for your support**
> > >
> > > Thanks for your support! We appreciate your openness in sharing your remaining concern, and we are glad that our additional experiments have effectively addressed it.
> > >
> > > Your follow-up comments prompted us to explore this direction more deeply, which in turn enabled us to clearly establish the broader utility of our benchmark for improving VLM robustness to multimodal deception. We are sincerely grateful for your constructive feedback.

---

### Official Review · Reviewer_cLyQ · 2025-11-01

**Soundness:** 3
**Presentation:** 3
**Contribution:** 3
**Rating:** 6
**Confidence:** 2

**Summary:**

This paper introduces DECEPTIONDECODED, a large-scale benchmark for understanding and detecting misleading creator intent in multimodal news. This work centers on modeling the combination of desired influence and execution plan behind deceptive news creation. The benchmark comprises 12,000 image–caption–article triplets, each grounded in trustworthy news contexts from VisualNews and simulated through intent-guided generation using GPT-4o and FLUX.1. It supports three intent-centric tasks: (1) misleading intent detection, (2) misleading source attribution, and (3) creator desire inference. Comprehensive evaluations of 14 VLMs reveal that even leading models struggle to reason about creator intent. Fine-tuning on DECEPTIONDECODED improves performance on external MMD benchmarks (e.g., MMFakeBench), underscoring its transferability.

**Strengths:**

1.The shift from content-level to intent-level misinformation detection is a underexplored direction that extends beyond factual correctness to communicative objectives.
2.The benchmark construction pipeline is well thought out: grounded in verified contexts, controlled through creator-intent configurations, and validated via human evaluation with high inter-annotator agreement
3.Evaluating 14 diverse VLMs across multiple reasoning paradigms (implication vs. consistency-oriented) offers valuable cross-model insights into weaknesses in current multimodal reasoning.

**Weaknesses:**

1.Absence of joint multimodal deception synthesis. Misleading instances are generated by modifying either text or image, but never both simultaneously. This simplification limits the ecological realism and potential difficulty of the dataset, as real-world misinformation often involves coordinated visual–textual deception.
2.Conceptual ambiguity of AI synthesis traces in the news context. The authors explicitly exclude AI-generated artifacts from being considered evidence of deception. However, this assumption may be problematic in the journalism setting, since the very use of AI-generated visuals in news reporting could itself indicate misleading or inauthentic intent.
3.Over-reliance on generative synthesis instead of precise editing. The misleading visuals are produced entirely through image generation (e.g., FLUX.1), without exploring fine-grained editing operations such as subtle facial expression shifts, contextual object replacements, or localized semantic changes. This may reduce the dataset’s ability to capture nuanced real-world manipulations.

**Questions:**

1.Have the authors compared DECEPTIONDECODED against existing unimodal intent detection models? Can the dataset still challenge models when only one modality is provided? This would clarify the necessity and distinct value of multimodal intent reasoning.
2.Could the framework be extended to include dual-modality misleading constructions. For instance, simultaneously altering both caption and image? Would such jointly deceptive instances better reflect real misinformation and raise the benchmark’s difficulty?
3.Have the authors quantified the number of samples for each misleading-intent category or verified that the dataset is balanced across intent types and news domains? If certain intents (e.g., political or health-related deception) dominate, how might this imbalance affect model evaluation and the generality of conclusions about “intent reasoning” performance?

---

> ### Author Response · Authors · 2025-11-23
> **Rebuttal [1/5]**
>
> Thanks for your feedback. We’re pleased that you found our intent-oriented perspective novel, our benchmark construction pipeline well-designed and validated, our experiments comprehensive, and our insights valuable.
>
> We hope our point-by-point responses addresses your concerns:
>
> ---------------------
>
> **Weakness 1: Misleading instances are generated by modifying either text or image, but never both simultaneously. This simplification limits the ecological realism and potential difficulty of the dataset, as real-world misinformation often involves coordinated visual–textual deception.**
>
> **Answer:** We appreciate the reviewer's insight regarding the importance of simultaneous visual textual deception for **ecological realism**. We agree that real-world misinformation can often involve fully fabricated or factually inconsistent text and images.
>
> Our work, however, is strategically focused on isolating and studying a **distinct and highly common threat model: implication-level deception** achieved through subtle manipulation of only a single modality. This focus is justified by both psychological evidence and the rigorous challenge it poses to VLM reasoning.
>
> Our core design choice is based on the reality that much real-world misinformation arises from the **strategic reframing of genuine content** rather than fully fabricating new stories (Lines 43-71 of submission). As supported by studies [1, 2], small edits to a single component (e.g., the image or the caption) can significantly **alter readers' memory, inferences, and intentions**, even when the underlying event remains factually unchanged.
>
> In our benchmark, **the deception remains coordinated** but is based on implication: **the unmodified modality lends credibility** to the one that has been subtly manipulated. The resulting image-caption pair appears factually consistent and grounded in a trustworthy source, but the subtle change subtly shifts the reader's interpretation (as illustrated in **Figure 1**). This makes the deception harder to detect than explicit multimodal factual conflict.
>
> Our experiments (Section 4) confirm that **restricting the manipulation to a single modality does not make the task trivial**. State-of-the-art models still struggle in this regime, demonstrating that distinguishing between subtle shifts in implication and authentic content requires deep semantic reasoning that current VLMs lack.
>
> We acknowledge that our benchmark does not cover cases where both modalities are simultaneously fabricated or made factually inconsistent, which we view as a complementary threat model that breaks factual alignment. Our contribution is to provide the first in-depth study of **intent-guided, single-modality manipulations** that are easily overlooked precisely because they avoid explicit multimodal conflict.
>
>
> We acknowledge that extending the benchmark to include dual-modality manipulations (perhaps by combining two subtle single-modality manipulations to compound the overall deception) is **a promising and necessary direction for future work** built upon the foundations established here.
>
>
> [1] The effects of subtle misinformation in news headlines. In Journal of Experimental Psychology: Applied, 2014.
>
> [2] Epistemic language in news headlines shapes readers’ perceptions of objectivity. In PNAS 2024.

---

> ### Author Response · Authors · 2025-11-23
> **Rebuttal [2/5]**
>
> **Weakness 2.  The authors explicitly exclude AI-generated artifacts from being considered evidence of deception. However, this assumption may be problematic in the journalism setting, since the very use of AI-generated visuals in news reporting could itself indicate misleading or inauthentic intent.**
>
> **Answer:** We appreciate the reviewer's observation. It highlights an important, evolving tension in journalism: the use of AI-generated content versus the presence of genuine misleading intent. We agree that in specific high-stakes contexts (e.g., deepfakes), AI-generated traces can signal deception.
>
> However, we maintain that for a robust, generalizable benchmark, the presence of AI-generated artifacts cannot be considered a reliable signal of misinformation. This is supported by evolving news industry practice and the design of our benchmark:
>
> **Decoupling Artifacts from Deceptive Intent**
>
> The rapid diffusion of generative tools has fundamentally changed newsroom workflows. As demonstrated by recent studies [3, 4], many reputable media outlets now legitimately employ AI-generated visuals for benign and well-defined roles, such as:
>
> * Conceptual illustrations and synthetic infographics.
>
> * AI-enhanced images to correct quality or lighting.
>
> * Synthetic portraits used to protect the identity of vulnerable sources.
>
> **In these legitimate uses, the images often contain visible synthesis signatures, yet their use is ethical and does not constitute misinformation**. To avoid conflating the production method with the communication outcome, we must exclude AI-trace detection as a primary signal of deception.
>
> **Validation through Benchmark Design**
>
> Our dataset's construction enforces this necessary decoupling (detailed in **Figure 2** and **Section 3.3**):
>
> * Our **Non-Misleading (Image)** samples deliberately include **AI-reconstructed or synthesized images** that accurately depict the news event.
>
> * Conversely, our **Misleading (Text)** samples use **entirely authentic photographs** with only the text modified to deceive.
>
> In both cases, the production method alone fails to reveal whether the content distorts the underlying event.
>
> **Our benchmark evaluates whether the content, regardless of its creation method, misrepresents context or meaning.** This design aligns with the practical needs of journalism and content moderation, which require interpreting **claims, implications, and contextual distortions** (the what and why), rather than simply flagging the presence of generative artifacts (the how).
>
> By excluding artifact detection, we ensure DeceptionDecoded provides a genuine test of a VLM's ability to detect intent-driven semantic manipulation, which is the core challenge of future misinformation detection.
>
>
> [3] Generative Visual AI in News Organizations: Challenges, Opportunities, Perceptions, and Policies. In Digital Journalism, 2024.
>
> [4] Guiding the way: a comprehensive examination of AI guidelines in global media. In AI&Society, 2025.

---

> ### Author Response · Authors · 2025-11-23
> **Rebuttal [3/5]**
>
> **Weakness 3.  Over-reliance on generative synthesis instead of precise editing. The misleading visuals are produced entirely through image generation (e.g., FLUX.1), without exploring fine-grained editing operations such as subtle facial expression shifts, contextual object replacements, or localized semantic changes. This may reduce the dataset’s ability to capture nuanced real-world manipulations.**
>
> **Answer:** We sincerely thank the reviewer for highlighting the importance of incorporating **precise image editing operations** to capture nuanced real-world manipulations. We fully agree that as powerful image editing models (such as Google’s Nano Banana) emerge, fine-grained edits, such as subtle facial expression shifts, contextual object replacements, or localized semantic changes, represent a challenging and ecologically valid risk.
>
> To address this, we conducted a new pilot experiment that applies our intent-guided deception framework directly to high-fidelity image editing rather than full image synthesis.
>
> **The construction pipeline follows our existing setup (Section 3.1 and Section 3.2):** we first establish the creator’s communicative intent (desired influence and execution plan), and then generate detailed image editing instructions aligned with that intent.
>
> * Editing Instructions: The LLM generates specific, localized instructions for the editing model, for example: *Using the provided image, please [add/remove/modify] [element]. Ensure the change is [integration details].*
>
> * Editing Model: We employed Google’s Nano Banana model, a state-of-the-art VLM capable of generating high-fidelity, locally coherent edits.
>
> * Pilot Dataset: Following the setup in Section 5.2, we created 400 new edited samples (200 for $\text{M-Sub (Image)}$ and 200 for $\text{M-Sig (Image)}$) based on a pilot set of 200 source news items.
>
> Using the same consistency-oriented detection prompt (Figure 15) as our main experiments, we benchmarked five representative VLMs. The results are summarized below:
>
> | Model           | M-Sub (Image) GPT-gen | M-Sub (Image) Edited | M-Sig (Image) GPT-gen | M-Sig (Image) Edited |
> |----------------|------------------------|-----------------------|------------------------|-----------------------|
> | GPT-4o         | 66.0                   | 48.0                  | 84.0                   | 61.0                  |
> | GPT-4o-mini    | 47.5                   | 22.5                  | 61.5                   | 41.5                  |
> | Claude-3.5-Haiku | 9.0                  | 6.0                   | 18.0                   | 11.5                  |
> | Qwen2.5-72B    | 53.5                   | 32.0                  | 57.0                   | 52.5                  |
> | Qwen2.5-32B    | 37.0                   | 21.5                  | 46.0                   | 41.0                  |
>
>
> The results yield three key observations that confirm the practical relevance of our approach:
>
> 1. **Increased Challenge**:  VLMs are consistently and noticeably weaker at detecting visually conveyed deception in the precisely edited samples compared to the fully generated (GPT-Gen) images.
>
> 2. **Generalization of our Simulation Framework**: The heightened difficulty demonstrates that our **intent-guided simulation framework naturally extends to diverse, realistic manipulation types** (including fine-grained editing), confirming the versatility of our conceptual approach.
>
> 3. **Risk Highlighted**: We also found that the image editing model could bypass established safety guardrails, producing intent-aligned manipulations at near-perfect success rates.
>
> These findings highlight the broader utility of our intent-guided framework in simulating multimodal misinformation, and reinforce heightened risks posed by subtle, intent-driven visual manipulation.
>
> **We have integrated these experiments, including illustrative examples of the editing setup, into Section 5.3 of the revised manuscript.**

---

> ### Author Response · Authors · 2025-11-23
> **Rebuttal [4/5]**
>
> **Question 1.  Have the authors compared DECEPTIONDECODED against existing unimodal intent detection models? Can the dataset still challenge models when only one modality is provided? This would clarify the necessity and distinct value of multimodal intent reasoning. **
>
> **Answer:** Thanks for raising this key question regarding the necessity and distinct value of multimodal intent reasoning compared to unimodal detection. Our analysis confirms that DeceptionDecoded remains highly challenging in unimodal settings, and further reveals a significant weakness in current models when integrating the visual modality.
>
> **Unimodal Challenge Validation**
>
> Although DeceptionDecoded is designed for multimodal benchmarking, our primary evaluation suite (**Table 2**) includes strong closed-source VLMs (such as Claude-3.7-Sonnet and Gemini-2.5-Pro) that are highly capable on language-only tasks. Their sub-optimal performance on samples with misleading captions confirms that detecting our intent-guided textual manipulation is non-trivial.
>
> More importantly, **Section 4.3** empirically isolates the unimodal textual reasoning task by comparing two settings where the misleading intent is embedded in the caption:
>
> * **T+A (Text + Article)**: A **unimodal textual reasoning setup** comparing the manipulated caption against the trustworthy article.
>
> * **Full (Image + Text + Article):** The complete multimodal setting.
>
> The relevant part of Table 4 is copied below for convenience:
>
> | **Model**       | **M-Sub (Text)** T+A | **M-Sub (Text)** Full | **M-Sig (Text)** T+A | **M-Sig (Text)** Full |
> |-----------------|-----------------------|------------------------|-----------------------|------------------------|
> | GPT-4o          | 63.4                  | **70.2**               | 88.4                  | **93.3**               |
> | GPT-4o-mini     | **36.8**              | 18.8                   | **64.4**              | 45.9                   |
> | Qwen2.5-VL-72B     | **53.6**              | 49.6                   | **83.8**              | 82.5                   |
> | Qwen2.5-VL-32B     | **35.8**              | 27.5                   | **67.2**              | 58.6                   |
>
> These results demonstrate that even in the unimodal **T+A** condition, performance is far from solved, indicating that the intent-guided textual manipulation alone is a significant challenge.
>
> **Empirical Support for Multimodal Distinct Value**
>
> The distinct value of the multimodal task is proven by the striking failure mode observed when the image is introduced: Several highly capable models (GPT-4o-mini, Qwen2.5-VL-72B, Qwen2.5-VL-32B) **perform worse in the full multimodal setting than in the unimodal T+A setting.**
>
> This performance drop occurs because the image provided is **factually correct** and aligns visually with the manipulated caption, strengthening the **surface-level image-text coherence**. Models often **over-trust this visual alignment** and are consequently distracted from identifying the deceptive framing in the caption.
>
> Together, these findings strongly support the **necessity** of the multimodal benchmark. DeceptionDecoded not only confirms that unimodal intent detection is hard, but it also reveals a critical weakness in current VLMs: their tendency to prioritize spurious cross-modal coherence over deep, evidence-based reasoning, confirming that full multimodal intent detection requires capabilities that current systems still lack.
>
> —--------------------------
> **Question 2.  Could the framework be extended to include dual-modality misleading constructions. For instance, simultaneously altering both caption and image? Would such jointly deceptive instances better reflect real misinformation and raise the benchmark’s difficulty?**
>
> **Answer:** Please refer to our **answer to Weakness 3.**

---

> ### Author Response · Authors · 2025-11-23
> **Rebuttal [5/5]**
>
> **Question 3. Have the authors quantified the number of samples for each misleading-intent category or verified that the dataset is balanced across intent types and news domains? If certain intents (e.g., political or health-related deception) dominate, how might this imbalance affect model evaluation and the generality of conclusions about “intent reasoning” performance?
>
> **Answer:** Thanks for raising this point. We have ensured that the dataset is balanced across news domains while allowing the distribution of specific misleading intents to reflect real-world salience.
>
> **Balance Across News Domains**
>
> We confirm that **DeceptionDecoded is constructed to be fundamentally balanced across its source material**. We selected 10 broad news domains (spanning politics, public health, disasters, economics, etc.) based on their high societal relevance.
>
> * We sample **200 high-quality, trustworthy news items** from each of the 10 domains. (**Section 3.1, Line 206**)
>
> * We construct six variants per item (three misleading, three non-misleading).
>
> This guarantees that **no single domain dominates** the content, preventing models from exploiting simple topical or domain cues.
> **Quantification of Misleading Intent Categories**
>
> While the source domains are balanced, the distribution of the misleading-intent categories (defined by "desired influence" in Section 3.2) is intentionally varied to reflect observed threat patterns. Across all 12,000 samples in DeceptionDecoded, we summarize:
>
> | Intent Type                            | M-Sub (Text) | M-Sig (Text) | M-Sub (Image) | M-Sig (Image) |
> | -------------------------------------- | ----- | ----- | ----- | ----- |
> | Political Polarization                 | 581   | 732   | 462   | 566   |
> | Social Polarization                    | 467   | 517   | 490   | 519   |
> | Cultural and Religious Polarization    | 29    | 27    | 19    | 9     |
> | Economic Misinformation                | 309   | 268   | 272   | 191   |
> | Public Health and Safety               | 913   | 942   | 1,011 | 1,096 |
> | Environmental and Scientific Polarization    | 467   | 430   | 531   | 509   |
> | Geopolitical and International Relations | 319   | 501   | 305   | 390   |
> | Psychological and Emotional Manipulation  | 1,744 | 1,911 | 1,886 | 1,944 |
>
> Three categories: “Psychological and Emotional Manipulation”, “Public Health and Safety”, and “Political Polarization” are the most frequent. This distribution is highly consistent with findings from extensive propaganda and misinformation research [5, 6]. Modern influence campaigns rely heavily on **emotional activation** and disproportionately target high-salience domains like **health and politics**.
>
> The category Cultural and Religious Polarization, while present, appears less frequently. This is a result of our ethical sourcing and sampling (Section 3.1 and Appendix B). We avoid synthetic oversampling of sensitive content to prevent reinforcing stereotypes or introducing bias.
>
> Notably, the fundamental domain balance (all 10 domains contribute equally) and the fact that each intent type appears across multiple domains and modalities ensures that models cannot rely on simple topic-level priors. Performance differences thus reflect the VLM's ability to reason about implication-level distortions and the structure of the deceptive intent, ensuring the generality of conclusions about "intent reasoning" performance.
>
> **We have included this detailed distribution and analysis in Appendix E.3 of the revised manuscript (highlighted in blue) to make the dataset structure more explicit.**
>
>
> [5] Mind Over Media: Propaganda Education for a Digital Age. Renee Hobbs, 2020.
>
> [6] The disaster of misinformation: a review of research in social media. In International Journal of Data Science and Analytics, 2022.

---

> ### Author Response · Authors · 2025-11-27
> **Gentle reminder of the author-reviewer discussion deadline**
>
> Dear Reviewer cLyQ:
>
> We are eager to engage in further discussions with you! In our earlier responses, we have actively addressed your concerns by providing:
>
> 1. Explanations on the **ecological realism and coordinated difficulty** of DeceptionDecoded
>
> 2. Clarifications on **why AI-generated visuals alone does not imply misleading or inauthentic intent**
>
> 3. An extension of our intent-guided misinformation simulation framework to **precise editing**
>
> 4. Empirical validation of the **distinct value of multimodal intent reasoning**
>
> 5. **Analysis of intent type distribution** in DeceptionDecoded
>
> As the author-reviewer discussion deadline is approaching, please let us know if you have any additional questions or concerns about the paper. We would be delighted to continue the conversation with you.
>
> We sincerely hope that our responses have effectively addressed your concerns and may encourage a more favorable reconsideration of our paper.
>
> Best regards,
>
> Submission 1711 Authors

---

### Author Response · Authors · 2025-12-03
**General Response [1/2]: Rebuttal Summary for the New AC**

Dear AC,

We sincerely appreciate your willingness to step in during this unprecedented review cycle. We understand the workload this entails and have summarized our rebuttal and key contributions below to assist your decision-making.

## Consensus on Strengths

Despite the mixed initial scores (6/4/4/4), all reviewers reached a consensus on the fundamental value and rigor of our work. They explicitly acknowledged:

* **Novelty & Timeliness**: The shift from content-level to **intent-level misinformation detection** is "underexplored" and "timely" (Reviewers cLyQ, cUUE, f8Fr).

* **Methodological Rigor**: Our **intent-guided news simulation framework** is viewed as theoretically grounded, well-controlled, and validated by human evaluation (Reviewers cLyQ, cUUE, fbiN).

* **Comprehensive Evaluation**: Our benchmarking of **14 VLMs** across diverse reasoning modes, as well as the **insights on key VLM vulnerabilities**, are thorough and valuable (All reviewers).

* **Broader Utility of Intent-Guided News Simulation**: Beyond diagnostic contributions, fine-tuning VLMs on DeceptionDecoded demonstrates **strong generalization** to real-world multimodal misinformation detection tasks (Reviewers cUUE, fbiN).


## Rebuttal Highlights & Score Improvement

We are especially encouraged by the constructive dialogue with **Reviewer cUUE (initial score: 4)**. Following our extended experiments on transferability, Reviewer cUUE acknowledged the broader impact of our work, stating **willingness to raise to a positive score**: **"The model's transferability is surprisingly impressive, and I'm inclined to raise the score."**

We have provided comprehensive responses and new experiments addressing all concerns from Reviewers cLyQ, f8Fr, and fbiN. While we unfortunately did not receive follow-up comments from them, we believe our extensive clarifications and additional experiments fully resolve their initial reservations.
To summarize the resolved issues for each reviewer:

* **Reviewer cLyQ:** We addressed ecological realism by explaining the **coordinated difficulty** and showing how **AI-generated visuals do not imply misleading intent**. We provided empirical validation of the **distinct value of multimodal intent reasoning** and extended our framework to **precise image editing**.

* **Reviewer f8Fr:** We clarified methodological integrity by verifying the **diversity of generated intents** and the **clear boundary between subtle/significant manipulations**. We provided the **conceptual distinction and implications of our two reasoning paradigms** (implication- and consistency-oriented) and reinforced our commitment to **ethical considerations** and appropriate **evaluation metric selection** given the data's class balance.

* **Reviewer fbiN:** We substantiated the dataset's quality by explaining why **potential generator-specific artifacts do not influence evaluation** and provided **evidence that style-related bias was removed** in our reframing experiment. We offered insights into **human disagreement (edge cases)** and clarified the **central role of news articles as necessary evidence**. Finally, we explained the models' **trivial behavior (the "NM Bias")** and analyzed **why the external framing hint affects accuracy** so dramatically.

We hope this summary assists with the remainder of the process, and remain confident that our contribution through investigating multimodal misinformation through the novel intent-based lens provides a valuable asset for future misinformation research.

Sincerely,

Submission1711 Authors

---

> ### Author Response · Authors · 2025-12-03
> **General Response [2/2]: Summary of Key Refinements in Updated Manuscript**
>
> Benefiting from reviewer feedback, we have strengthened our manuscript with four key refinements:
>
> **1. Validation of Ecological Validity (Image Editing)**
> * **Reviewer Feedback**: *Does the intent-guided simulation framework apply to realistic, fine-grained image editing?*
> * **Refinement:** We extended our framework to guide **high-fidelity image editing** using Google's Nano Banana model (**Section 5.3**).
> * **Result:** The framework successfully generates subtle, localized manipulations. Crucially, we found that fine-grained edits are even harder for VLMs to detect than full synthesis, reinforcing the urgency of our benchmark.
>
> **2. Proof of Methodological Utility (Generalization)**
> * **Reviewer Feedback**: *Does the synthetic data help models detect real multimodal misinformation?*
> * **Refinement**: Inspired by Reviewer cUUE, we extended transferability experiments in **Section 5.1** to two additional popular real-world benchmarks (**Fakeddit** and **FakeNewsNet**).
> * **Result**: Fine-tuning on our data yields massive gains (up to **+30.7% Macro-F1**) across independent real-world benchmarks. This proves our framework also serves as a **concrete method for enhancing VLM robustness**, beyond providing valuable diagnostic insights.
>
> **3. Substantiation of Semantic Diversity in DeceptionDecoded**
> * **Reviewer Feedback**: *Is the intent diversity sufficient in DeceptionDecoded?*
> * **Refinement**: We provided a detailed distribution analysis (**Appendix E.3**) showing that our **combinatorial intent framework** (through formulating desired influence and execution plan) ensures **presence of each intent type across all four manipulation categories** (subtle/signature, text/visual). This confirms that intents are expressed in varied modalities and transformation styles. Furthermore, the intent type distribution mirrors real-world misinformation patterns, validating that our framework is **following real-world salience** in establishing creator intents.
>
> **4. Analysis of Edge Cases in Human Evaluation**
> * **Reviewer Feedback**: *Why do human annotators disagree on the label of some samples (misleading vs. non-misleading)?*
> * **Refinement**: As detailed in **Appendix F.1** and **Figures 7-10**, we analyzed all disagreement cases and found 91.7% stem from **intent ambiguity**: cases where the manipulation operates at the threshold of perception. This validates that our benchmark successfully captures the subtlety of **implicit deception**, rather than simple annotation noise.

---

### Meta-Review · Area_Chair_tepJ · 2025-12-25

**Summary:**

From reviews and feedback, the score might change from a "borderline reject" to a "borderline accept".
This paper introduces DECEPTIONDECODED, reframing multimodal misinformation detection from surface-level matching to intent-level (implication-level) deception, where content is technically true but strategically misleading. The authors propose a structured framework of creator goals and execution strategies and release a 12K-example dataset grounded in real, trustworthy news with subtle manipulations.

Reviewers found the task formulation timely and realistic, and the dataset a strong benchmark that exposes consistent weaknesses in current vision-language models, particularly their over-reliance on fluent language and superficial consistency. Evaluation across 14 models provides clear diagnostic insights.

While reviewers were initially concerned that the paper only identified problems without solving them, the authors' rebuttal provided new evidence that training models on this dataset significantly boosts their ability to detect real-world fake news on other platforms. Additional experiments with more advanced image edits further support the generality of the framework. I strongly encourage the authors to incorporate the new experiments and discussions from the rebuttal into the revised paper.

Overall, this paper is a well-motivated and has impactful contribution. Given that the paper's strengths outweigh the initial concerns, I am pleased to recommend acceptance.

**Reviewer Concerns:**

Concerns Effectively Addressed:

- (Concern from reviewer cLyQ) The authors successfully argued that the Intent-Guided Framework is itself a constructive method. They provided new empirical evidence showing that fine-tuning on their benchmark DECEPTIONDECODED yields gains on external datasets.

- (Concern from reviewer cUUE) The authors proved the generalizability of the framework to more nuanced visual manipulations.

Outstanding or Partially Unresolved Concerns:

- Despite high inter-annotator agreement, the "Intent Ambiguity" in subtle cases remains inherent (reviewer cUUE). The authors explained this well in the rebuttal, but it remains a boundary condition of the task.

- As noted by Reviewer fbiN, the benchmark uses "static" intent. The authors correctly identified that capturing dynamic sociopolitical incentives and audience feedback loops is the "next frontier" and currently out of scope.

**Reviewer Scores:**

Reviewer cLyQ is likely to keep their score at 6. Reviewer cUUE has indicated they will raise their score, likely to 6. Reviewers f8Fr and fbiN may keep their scores at 4, or could increase them to 6.

---

### Decision · Program_Chairs · 2026-01-26

Accept (Poster)